# DNA damage induced by HIV-1 Vpr triggers epigenetic remodeling and transcriptional programs to enhance virus transcription and latency reactivation

Nicholas Saladino[1], Emily Leavitt[1], Hoi Tong Wong[2], Jae-Hoon Ji[3], Diako Ebrahimi[4], Daniel J. Salamango [1]*

1 Department of Microbiology, Immunology, and Molecular Genetics, The University of Texas Health Science Center at San Antonio, San Antonio, Texas, United States of America, 2 Department of Microbiology, Mt. Sinai School of Medicine: Icahn School of Medicine, New York, New York, United States of America, 3 Department of Biochemistry and Structural Biology and Greehey Children's Cancer Research Institute, The University of Texas Health Science Center at San Antonio, San Antonio, Texas, United States of America, 4 Texas Biomedical Research Institute, San Antonio, Texas, United States of America

* salamango@uthscsa.edu

## Abstract

Hijacking of host DNA damage repair (DDR) pathways to facilitate virus replication is broadly conserved amongst diverse viral families. It has been well established that the HIV-1 accessory protein Vpr induces constitutive DDR signaling and G2/M cell cycle arrest, but the virologic function of this activity remains unclear. Here, we use a combination of functional, pharmacologic, biochemical, and genetic approaches to establish that virion-associated and *de novo* Vpr proteins induce DDR responses that trigger global epigenetic remodeling and activation of transcription programs to enhance HIV-1 promoter activity during acute infection and reactivation from latency. Functional, genetic, and bimolecular fluorescence complementation experiments reveal that Vpr segregates into two functionally discrete pools—a multimeric pool in the nucleus associated with chromatin and a monomeric pool in the cytoplasm associated with a host E3-ubiquitin ligase. Vpr-induced DDR and epigenetic remodeling activities are present in common HIV-1 subtypes circulating globally and in patient-derived isolates.

## Introduction

HIV-1 encodes four accessory proteins that are essential for virus replication *in vivo* [1–3]. The Vif, Vpu, and Nef accessory proteins promote virus replication by directly counteracting distinct host innate immune defense mechanisms, whereas the primary proviral function of the Vpr accessory protein remains enigmatic [4–6]. Vpr is highly conserved amongst primate lentiviruses and is critical for virus replication in myeloid

**Data availability statement:** All relevant data are within the paper, the Supporting information files, and the Figshare Data Repository (https://doi.org/10.6084/m9.figshare.c.8239897).

**Funding:** This work was supported by an NIAID R01 award AI189230 to DJS. It was also supported by startup funds from Stony Brook University and startup funds from the University of Texas Health Science Center at San Antonio. DJS, NS, EL, DE, and JJ received salary support from R01 AI189230. The funders had no role in study design, data collection and analysis, decision to publish, or preparation of the manuscript.

**Competing interests:** The authors have declared that no competing interests exist.

**Abbreviations**: BSA, bovine serum albumin; DDR, DNA damage repair; DTT, dithiothreitol; EDTA, ethylenediaminetetraacetic acid; MDMs, monocyte-derived macrophages; MFI, mean fluorescence intensity; PBS, phosphate-buffered saline; PFA, paraformaldehyde; PTMs, post-translational modifications; SDS, sodium dodecyl sulfate.

cells and for pathogenesis in vivo [7–11]. Canonically, Vpr engages a host CUL4/DDB1/DCAF1 E3-ubiquitin ligase complex to direct numerous cellular proteins for proteasomal degradation, resulting in systems-level remodeling of the host transcriptome and proteome [12]. Accumulating evidence indicates that a majority of these substrates are involved in DNA repair, DNA modification, or chromatin remodeling [12–14], which rationalizes observations that Vpr induces constitutive activation of ATM and ATR DNA damage repair (DDR) kinases, the accumulation of DNA strand breaks, and G2/M cell cycle arrest [12,15–18]. Additionally, recent studies revealed that Vif and Vpu independently antagonize diverse DDR pathways to block the activation of antiviral defenses triggered by abnormal DDR signaling or recognition of viral cDNA [19,20], signifying that HIV-1 utilizes discrete strategies to fine-tune host DDR responses.

Another widely observed Vpr activity is the ability to influence HIV-1 promoter activity in multiple primary and immortalized cell models. For instance, Vpr-deficient viruses exhibit decreased transcription in monocyte-derived macrophages (MDMs) and dendritic cells, which are key cell types for virus replication [9,21–24]. Likewise, Vpr expression leads to enhanced transcription of unintegrated viral DNA and reactivation of virus from latency [25–29]. However, the direct cause-and-effect mechanism(s) underlying Vpr's ability to modulate virus transcription remain poorly understood. Biochemical and transcriptional studies of HIV-1 infected U2OS and MDMs recently revealed that Vpr-induced DNA damage is correlated with increased NFκB-driven transcription, providing a mechanistic clue for Vpr-directed HIV-1 promoter modulation [18]. Moreover, DNA-PK has been implicated in directly phosphorylating RNA polymerase II to promote HIV-1 transcription initiation and elongation [30,31].

Detection of DNA damage by cellular factors triggers a plethora of dynamic local and global epigenetic post-translational modifications (PTMs) that regulate chromatin accessibility and the binding of DNA repair factors [32–36]. The majority of these modifications involve increased acetylation of lysine residues at the N-terminal tails of core histone proteins. Importantly, several DDR-induced histone PTMs have been associated with increased HIV-1 transcription or reactivation of virus from latency, and multiple lines of evidence have established that decreased acetylation of nucleosomes at the HIV-1 promoter induces latency while increased acetylation is associated with transcriptional activation [37–43]. Moreover, DDR signaling induces the activation of broad-spectrum transcription factors to up-regulate genes involved in stress responses, DNA repair, cell cycle control, and apoptosis [44,45]. These observations inspired our hypothesis that Vpr-directed DDR signaling promotes epigenetic remodeling and activation of transcription programs to enhance HIV transcription at the integration site.

Here, we establish that Vpr induces global epigenetic remodeling in several cell models, including primary MDMs, to enhance HIV-1 promoter activity during acute infection and latency reactivation. Our structure-guided mutagenesis studies demonstrate that Vpr uses a conserved network of electrostatic interactions to induce DDR signaling and epigenetic changes that facilitate enhanced HIV-1 promoter activity. This comprehensive mutational analysis in combination with chemical inhibitor

experiments point to an inseparable relationship between Vpr-induced DDR signaling and epigenetic remodeling activity, that is independent from Vpr-induced G2/M cell cycle arrest. Genetic, biochemical, and bifluorescence complementation studies demonstrate that Vpr utilizes both degradation-dependent and -independent mechanisms to induce epigenetic remodeling and HIV-1 promoter activation, suggesting that Vpr exists as two pools with discrete functions. Further, genetic and pharmacologic studies indicate that de novo and virion-associated Vpr proteins induce DDR signaling to promote phosphorylation of several transcription factors known to interact with the HIV-1 promoter in addition to RNA polymerase II phosphorylation. Finally, functional studies and bioinformatic analyses indicate that induction of DDR signaling and epigenetic remodeling activities are prevalent among common HIV-1 subtypes circulating globally.

## Results

### Virion-associated and *de novo* Vpr proteins induce global epigenetic remodeling through the DNA damage response

To test our hypothesis that Vpr alters the cellular epigenetic landscape, we infected cells with Vpr-proficient (Vpr$_{WT}$) or -deficient (neg) HIV$_{NL4-3}$ reporter viruses (CMV-driven *mCherry* in place of *nef*) and profiled 17 histone PTMs 48 hours post-infection (Fig 1A and 1B; S1 Data). Of note, the inclusion of mCherry allows for identification of infected cells using a low multiplicity of infection (*i.e.,* infection rates were less than 10% to achieve more physiologic Vpr expression). Additionally, Vif and Vpu were functionally inactivated because they have been implicated in modulating DDR responses through mechanisms independent of Vpr (detailed in experimental procedures) [19,20]. Remarkably, 14 of 17 epigenetic marks exhibited a significant increase in acetylation, phosphorylation, or methylation in differentiated THP1 cells (macrophage-like) infected with Vpr$_{WT}$ virus compared to control infected cells (Figs 1B and S1A; S1 Data). Infection of primary MDMs and HeLa cells recapitulated these observations, indicating this is not a cell-type-specific phenomenon (S1B, S1C, and S2 Figs; S1 Data; S1 Raw Images). Importantly, immunoblotting and flow cytometry experiments confirmed Vpr-induced epigenetic changes in differentiated THP1 and HeLa cells that persisted for ≥96 hours post-infection (Fig 1C–1E, S2B, and S2C Figs; S1 and S2 Data; S1 Raw Images). These observations collectively suggested that Vpr-induced epigenetic remodeling is not constrained to individual histone proteins or amino acid residues; therefore, subsequent experiments evaluate diverse histone PTMs to highlight the breadth of this Vpr activity.

Next, we sought to delineate the upstream mechanism(s) underlying Vpr-directed epigenetic remodeling. As described above, a longstanding observation is that Vpr induces constitutive DDR signaling in multiple cell types (*ex.,* Figs 1, S1B, S1C, and S2D–S2F). Given that activation of DNA repair is known to alter the epigenetic landscape, we postulated that Vpr's ability to induce DDR signaling is a prerequisite for epigenetic remodeling activity. To test this, we treated infected cells with caffeine to simultaneously inhibit ATM/ATR and assessed Vpr-induced changes to histone marks. As anticipated, caffeine treatment ablated Vpr's ability to induce phosphorylation of the DDR marker γH2A.X, and importantly, also ablated the increased abundance of all histone PTMs examined in immunofluorescence microscopy and immunoblotting experiments (Figs 2A–2C, and S3B; S1 Data; S1 Raw Images). Moreover, treatment of Vpr$_{WT}$-infected cells with ATM (AZD1390) or ATR (NU6027) inhibitors recapitulated these observations (Fig 2B, inhibitor specificity demonstrated in S2F Fig; S1 Data).

Because HIV-1 exhibits a high degree of genetic diversity, we assessed the breadth of this activity across major HIV subtypes circulating globally. Consensus Vpr sequences from subtypes A, C, D, and AE were cloned into a modified version of the HIV$_{NL4-3}$ provirus depicted in Fig 1A (amino acid alignment in S3A Fig). To avoid disrupting the *tat* open reading frame, we expressed consensus *vpr* subtypes from a CMV-driven *mCherry-T2A* expression cassette in place of *nef*, which permits independent mCherry and Vpr expression from the same mRNA [46]. HeLa cells were infected with the indicated Vpr subtypes and assayed for DDR activation and changes to histone marks. All subtypes were capable of inducing γH2A.X focus formation and increasing the abundance of multiple histone PTMs, which could be ablated by caffeine

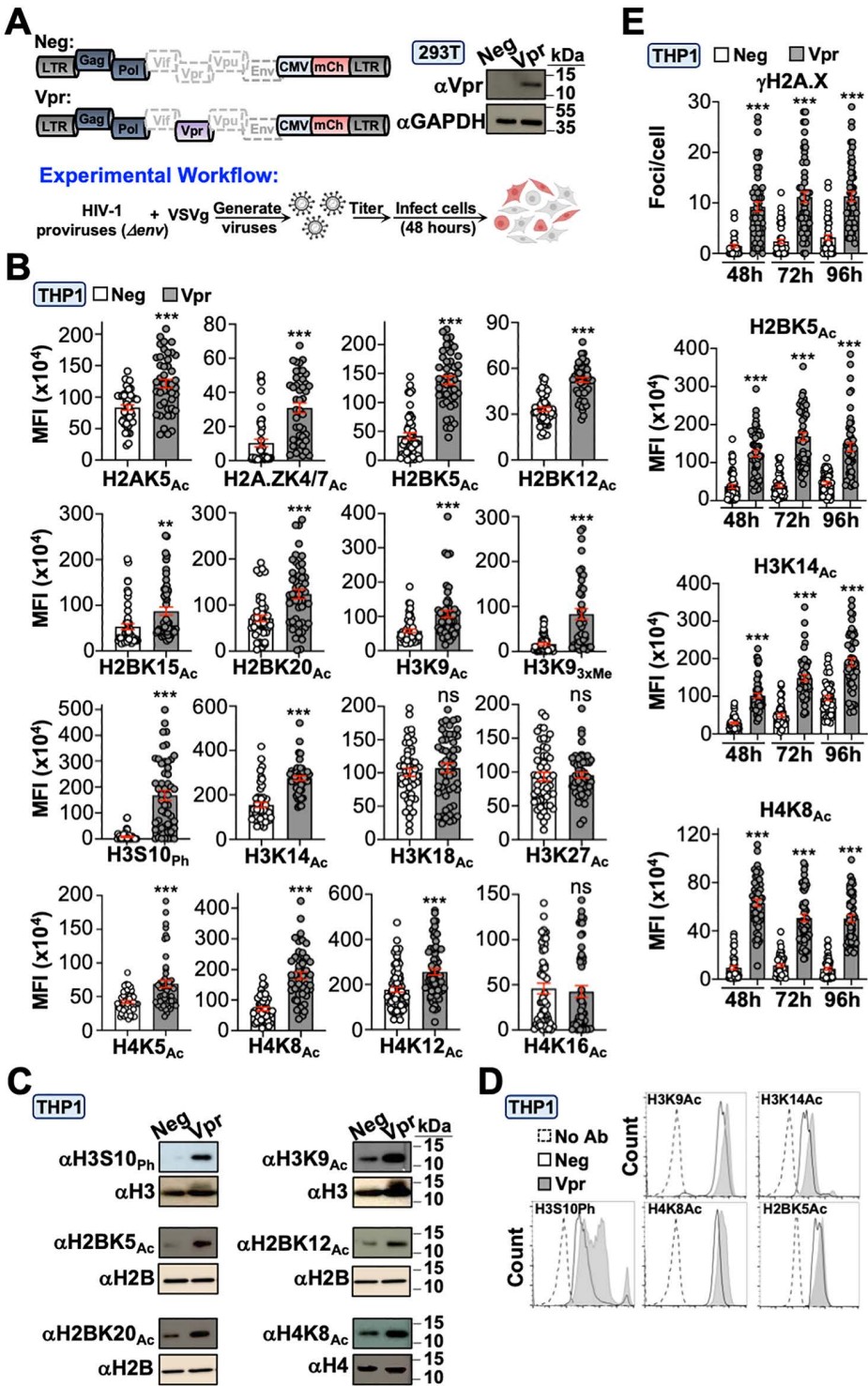

**Fig 1. Vpr induces global epigenetic remodeling. (A)** Schematic of our experimental workflow and validation of Vpr expression in HEK293T cells. HEK293T cells were infected for 48 hours with control or Vpr-expressing virus prior to preparation for immunoblot analysis. The unmodified images underlying this Figure can be found in S1 Raw Images. The BioRender license for the schematic is c65h229. For all experiments, cells were infected for 48 hours prior to analysis unless noted. Likewise, data presented are from one of three representative experiments unless noted. **(B)** Quantification of fluorescence microscopy images of histone marks in differentiated THP1 cells infected with control or Vpr-expressing virus ($n = 50$ cells). THP1

cells were simultaneously infected and treated with PMA (100 ng/mL) to induce differentiation for 48 hours prior to preparation for immunofluorescence microscopy. For quantification of immunofluorescence microscopy images, only mCherry-positive cells were considered for analysis. Images were obtained using an EVOS M5000 microscope and foci or mean-fluorescence were quantified using Image J software. For foci quantification, we used the "find maxima" imaging processing tool that quantifies foci above a preset threshold. Analyses performed using a student $t$ test: ns, not significant; *** $p < 0.001$; ** $p < 0.01$; * $p < 0.05$; and scale bar = 10 μM. The data underlying this Figure can be found in S1 Data. **(C)** Immunoblot analysis of histone marks in undifferentiated THP1 cells infected with control or Vpr-expressing virus 48 hours post-infection. Lysates were probed with antibodies against the specific histone mark indicated and the corresponding total histone protein was used as a loading control. The unmodified images underlying this Figure can be found in S1 Raw Images. **(D)** Intracellular immunolabeling and flow cytometric analysis of undifferentiated THP1 cells infected with control or Vpr-expressing virus 48 hours post-infection. Representative gating strategies are depicted in S2 Data and raw FSC files can be found in the Figshare Data repository (https://doi.org/10.6084/m9.figshare.c.8239897). **(E)** Time-course analysis of select histone marks in differentiated THP1 cells infected with control or Vpr-expressing virus ($n = 50$ cells). THP1 cells were simultaneously infected and treated with PMA (100 ng/mL) to induce differentiation for 48 hours prior to preparation for immunofluorescence microscopy. Analyses performed using a one-way ANOVA; *** $p < 0.001$. The data underlying this Figure can be found in S1 Data.

treatment (Fig 2C; S1 Data). Lastly, Vpr-induced activation of DDR signaling has been linked to G2/M cell cycle arrest [47–49]. While our observations in S1 Fig suggested that this Vpr activity is independent of cell cycle because primary MDMs are almost exclusively in G0/G1 (Fig 2D, left; S1 and S2 Data), we wanted to assess epigenetic changes in a cell model susceptible to Vpr-induced G2/M cell cycle arrest. HeLa cells were infected with Vpr or control viruses for 48 hours prior to cell cycle staining and flow cytometric analysis to quantify antibody labeling (mean fluorescence intensity [MFI]) at specific cell cycle stages. As anticipated, Vpr-driven DDR signaling and epigenetic changes were not correlated with G2/M cell cycle phase but rather with the G0/G1 phase, supporting our observations in primary MDMs (Figs S1 and 2D; S1 and S2 Data).

It is well established that Vpr is abundantly incorporated into nascent particles; however, recent studies have indicated that capsid disassembly likely occurs within the nucleus, making the functional role of virion-associated Vpr unclear. We speculated that virion-associated Vpr may induce DDR to alter the epigenetic landscape at the onset of acute infection. To test this idea, we treated differentiated THP1 or HeLa cells with either integrase (Raltegravir; Ral) or reverse-transcriptase (Etravirine, ETR, and Zidovudine, AZT) inhibitors and assessed the activation of DDR signaling. The hierarchy of DDR signaling requires the phosphorylation of H2A.X to promote the activation of ATM and ATR repair kinases, which then directly activate effector kinases CHK2 and CHK1, respectively. Pre-treatment with either Ral or ETR/AZT ablated infection in Vpr or control infected cells but not in vehicle-treated cells (Fig 3A; S1 Data). Importantly, HeLa or differentiated THP1 cells pre-treated with Ral or ETR/AZT exhibited a significant increase in phosphorylation of γH2A.X, pCHK1, and pCHK2 in Vpr-proficient virions but not in control virions (Fig 3A and 3B; S1 Data). Of note, ETR/AZT treatment also induced DDR signaling independent of Vpr expression which has been documented previously [50,51]. To further confirm these observations, we expressed Vpr in *trans* or utilized a provirus harboring a p6 mutation (FRFG-AAAA) defective for Vpr incorporation [52]. Infection of differentiated THP1 cells using either experimental approach resulted in the induction of γH2A.X foci for Vpr-proficient but not deficient virions (Figs 3C and 3D; S1 Data).

### Defining the Vpr surface required for DDR activation and epigenetic remodeling

The co-crystal structure of a Vpr/DCAF1/DDB1 ternary complex revealed that solvent-exposed Vpr surfaces exhibit a unique charge dichotomy with electronegative and electropositive N- and C-termini, respectively [53] (Fig 4A, PDB: 5JK7). Because of this, we wondered if the Vpr surface used for inducing DDR signaling and epigenetic remodeling would be electrostatic in nature. We utilized structure-guided mutagenesis to generate a large panel of charge-exchange single amino acid substitution mutations at Vpr surface residues separable from the *tat* open reading frame. Differentiated THP1 and HeLa cells were infected for 48 hours prior to immunofluorescence microscopy experiments to assess H2A.X and CHK1 phosphorylation (*i.e.,* activation of DDR signaling pathways, Fig 4B and 4C; S1 Data). This approach identified two regions of interest: the first encompassed N-terminal electronegative-to-electropositive substitutions that exhibited

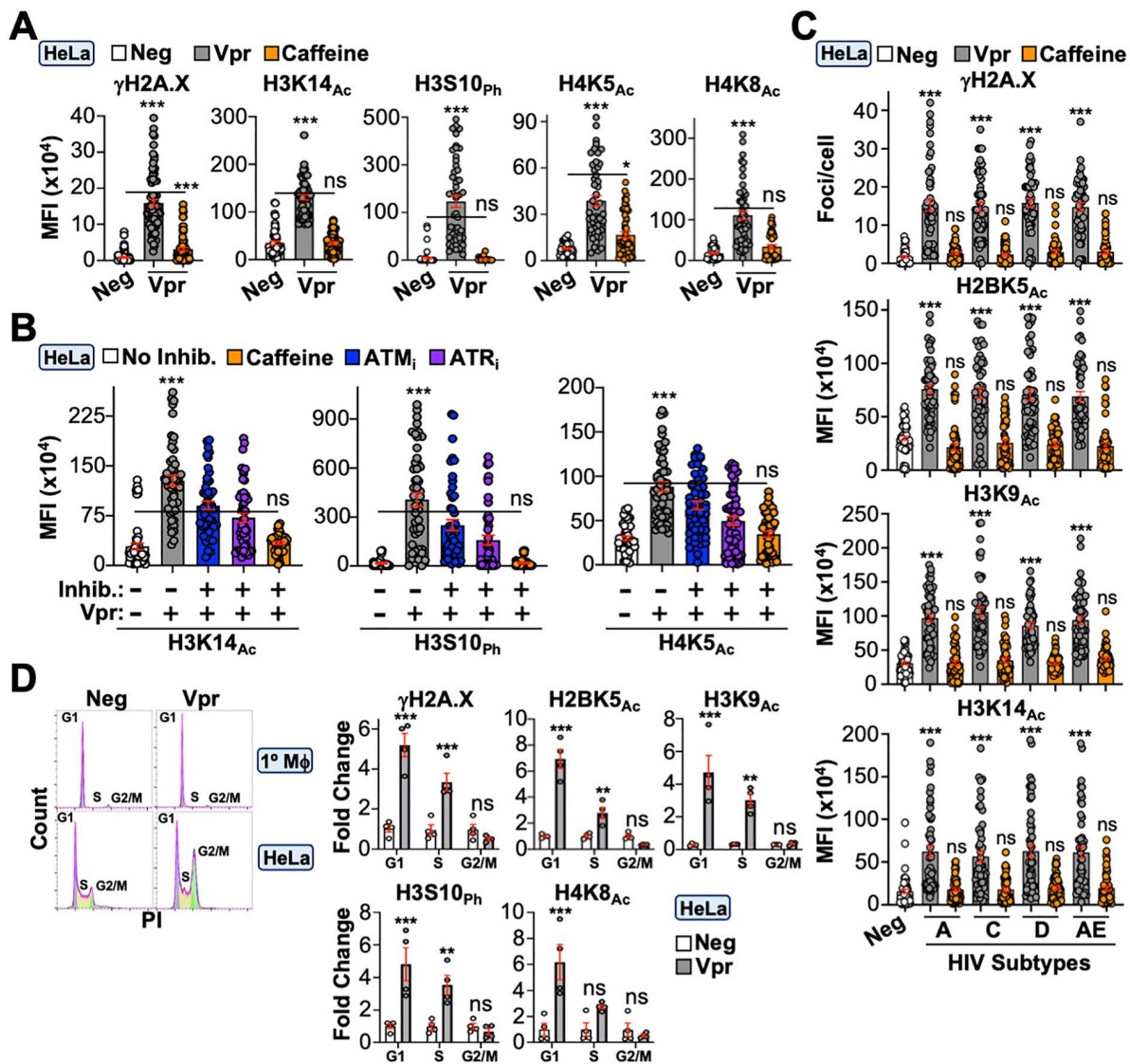

**Fig 2. Vpr induces epigenetic remodeling through the DNA damage response. (A)** Immunofluorescence microscopy quantification of DDR activation and histone marks in HeLa cells infected with control or Vpr-expressing virus. Vpr-infected HeLa cells were infected for 24 hours and then treated with vehicle or 3 mM caffeine for 24 hours ($n = 50$ cells). Analyses performed using a one-way ANOVA; ns, not significant; *** $p < 0.001$. The data underlying this Figure can be found in S1 Data. **(B)** Immunofluorescence microscopy quantification of histone marks in HeLa cells infected with control or Vpr-expressing virus in the presence or absence of vehicle, 3 mM caffeine, 10 nM ATM$_i$, or 10 μM ATR$_i$ ($n = 50$ cells). Cells were infected for 24 hours prior to inhibitor treatment for 24 hours and preparation for immunofluorescence microscopy. Analyses performed using a one-way ANOVA; ns, not significant; *** $p < 0.001$. The data underlying this Figure can be found in S1 Data. **(C)** Immunofluorescence microscopy quantification of histone marks in HeLa cells infected with control or virus expressing the indicated Vpr subtype consensus sequence. Cells were infected for 24 hours and then treated with vehicle or with 3 mM caffeine for 24 hours prior to preparation for immunofluorescence microscopy ($n = 50$ cells). Analyses performed using a one-way ANOVA; ns, not significant; *** $p < 0.001$. The data underlying this Figure can be found in S1 Data. **(D)** Cell cycle analysis of DDR activation and histone marks in HeLa cells infected with control or Vpr-expressing viruses. Left, flow cytometric analysis of propidium iodide (PI) stained HeLa or MDM cells infected with the indicated virus. Right, flow cytometric analysis of immunolabeled and PI-stained HeLa cells infected with the indicated virus ($n = 4$ experiment). Analyses performed using a one-way ANOVA; ns, not significant; *** $p < 0.001$; ** $p < 0.01$. The data underlying this Figure can be found in S1 Data. Representative gating strategies are depicted in S2 Data and raw FSC files can be found in the Figshare Data repository (https://doi.org/10.6084/m9.figshare.c.8239897).

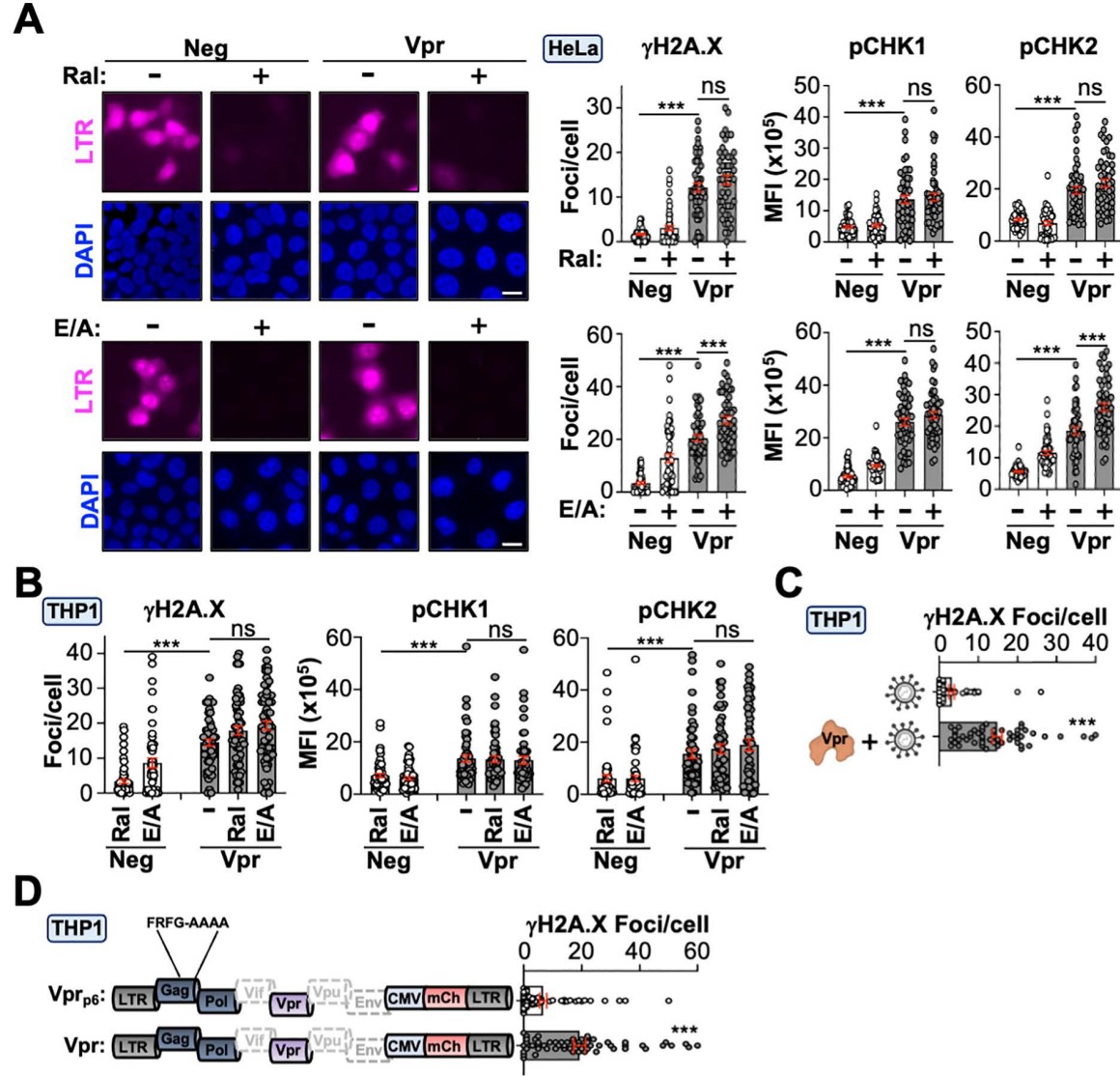

**Fig 3. Virion-associated Vpr proteins induce epigenetic remodeling. (A, B)** Representative fluorescence microscopy images and immunofluorescence quantification of DDR activation in HeLa and differentiated THP1 cells infected with Vpr or control viruses pre-treated with raltegravir or combination zidovudine/etravirine ($n = 50$ cells). For these experiments, the mCherry reporter is expressed from the viral promoter instead of being driven by CMV. This allows for tracking infection kinetics in real time to assess DDR activation from virion-associated Vpr. For raltegravir experiments, cells were pre-treated for 1 hour, infected with the indicated virus for 24 hours, and then prepared for immunofluorescence microscopy. For zidovudine/etravirine experiments, virus and drug were added at the same time and allowed to incubate with cells for 24 hours before analysis. Analyses performed using a one-way ANOVA; ns, not significant; *** $p < 0.001$. The data underlying this Figure can be found in S1 Data. **(C, D)** Immunofluorescence microscopy quantification of γH2A.X focus formation in differentiated THP1 cells infected with control virus or virus containing Vpr supplemented in trans (left), or with virus harboring a p6 mutation (FRFG-AAAA) that ablates Vpr incorporation into nascent virions (right) ($n = 50$ cells). Analyses performed using a student $t$ test; *** $p < 0.001$. The data underlying this Figure can be found in S1 Data.

increased activation of DDR signaling, while the second region involved C-terminal electropositive-to-electronegative substitutions deficient for DDR activation (Fig 4B and 4C; S1 Data). Importantly, hyperactive (Y15R) and loss-of-function (S79E) mutants exhibited an inextricable link between activation of DDR signaling and modulating histone PTMs in both immunofluorescence microscopy and immunoblotting experiments, which was not due to altered Vpr protein expression,

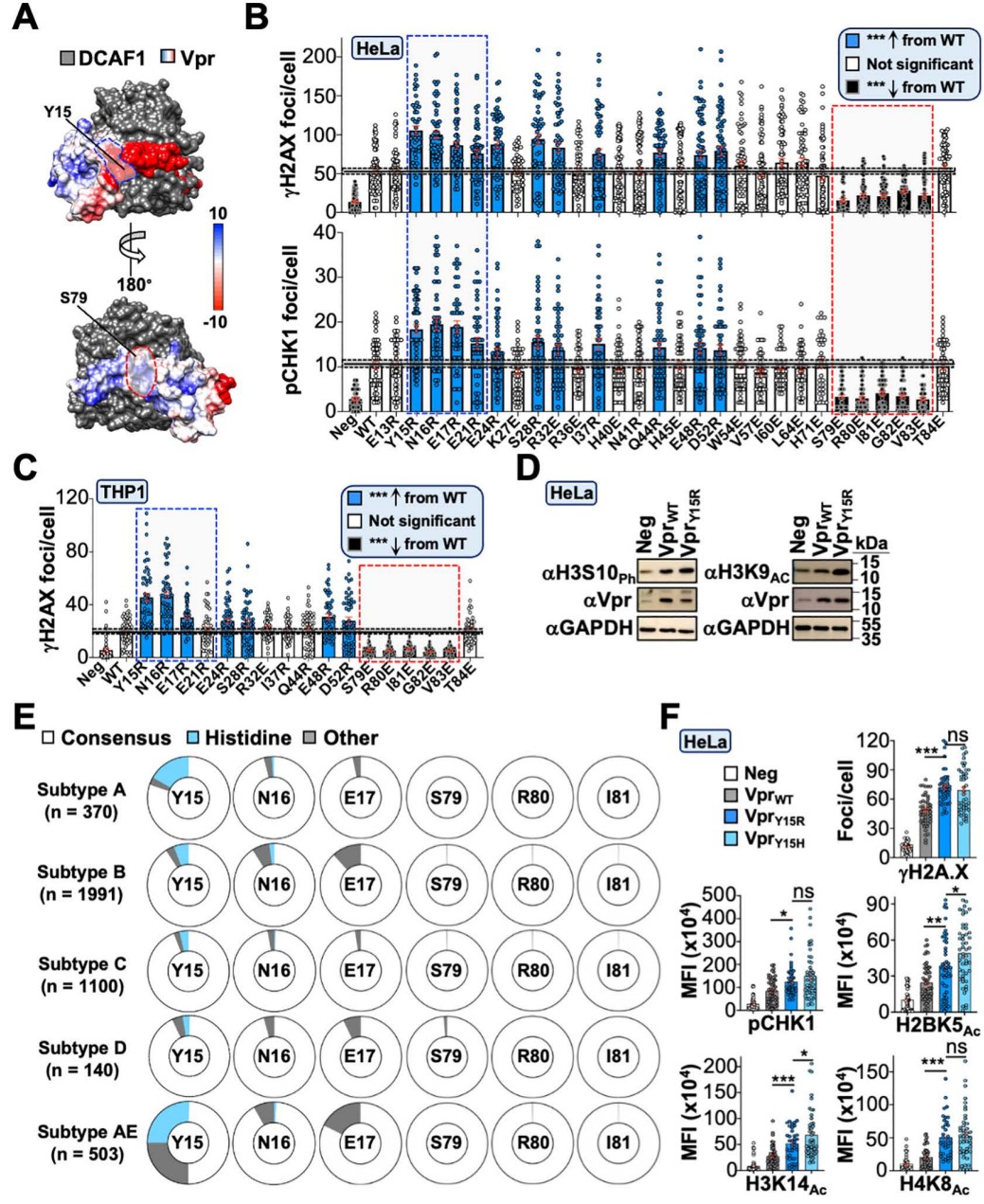

**Fig 4. Vpr surface residues used to induce DDR and histone marks.** (A) Electrostatic surface potential of a Vpr-DCAF1 co-complex. Surfaces associated with hyperactive or DDR-deficient Vpr mutants are colored in blue and red, respectively. (B, C) Immunofluorescence microscopy quantification of DDR activation following infection of HeLa or differentiated THP1 cells with Vpr_WT or mutant viruses 48 hours post-infection ($n = 50$ cells). Dashed boxes highlight hyperactive and DDR-deficient Vpr mutants associated with sub-surfaces colored blue and red, respectively, from **Fig 4A**. The data underlying this Figure can be found in S1 Data. (D) Immunoblot analysis of histone marks in HeLa cells infected with indicated viruses 48 hours post-infection. The unmodified images underlying this Figure can be found in S1 Raw Images. (E) Circle graphs displaying amino acid variance at relevant positions identified in the mutagenesis screen with consensus residues in the middle of the circle. Sequences obtained from Los Alamos database ($n \approx 4,100$). (F) Immunofluorescence microscopy quantification of DDR activation and histone marks following infection of HeLa cells with Vpr_WT, Vpr_Y15R, or Vpr_Y15H mutant viruses 48 hours post-infection ($n = 50$ cells). Analyses performed using a one-way ANOVA; ns, not significant; *** $p < 0.001$; ** $p < 0.01$; * $p < 0.05$. The data underlying this Figure can be found in S1 Data.

further indicating that Vpr-induced DDR signaling and epigenetic remodeling activity are correlated (Figs 4D, S3C, and S3D; S1 Data; S1 Raw Images).

We were intrigued by the hyperactive Vpr mutants and were curious if any were prevalent in patient-derived isolates. Using full-length HIV-1 sequences available in the Los Alamos database (>8,000 bp), we analyzed Vpr polymorphisms at key functional residues when corresponding subtype information was available (~4,100 sequences total). Our analyses indicated that Vpr residues S79, R80, and I81 were nearly ubiquitously conserved across all major Group M HIV-1 subtypes (Fig 4E), which supports functional studies in Fig 2C. We also assessed diversity at positions Y15, N16, and E17, and interestingly, the most abundant polymorphism among these residues was a histidine substitution at Y15 (Fig 4E). Because histidine mimics the positive charge exchange of the arginine substitution, we reasoned that Y15H would phenocopy Y15R. As anticipated, HeLa cells infected with $HIV_{NL4-3}$ carrying the $Vpr_{Y15H}$ mutation exhibited increased DDR signaling and histone PTMs compared to $Vpr_{WT}$ controls (Fig 4F; S1 Data).

## Vpr-directed DDR activation and epigenetic remodeling occur through DCAF1-dependent and -independent mechanisms

While the observations above unambiguously linked Vpr-induced DDR signaling with epigenetic remodeling activity, the molecular mechanism(s) driving these responses were still unclear. Because the best characterized Vpr function is the depletion of host proteins through a CUL4/DDB1/DCAF1 E3-ubiquitin ligase complex [12], we wanted to determine if E3-engagement is required for inducing DDR signaling and histone PTMs. Proviruses encoding Vpr mutants Q65R or H71R, which are deficient for DCAF1 binding [49,54], were generated and used to infect differentiated THP1 or HeLa cells. Surprisingly, most histone PTMs only exhibited an ~50% reduction following infection with $Vpr_{Q65R}$ or $Vpr_{H71R}$ viruses compared to $Vpr_{WT}$, suggesting that Vpr utilizes both DCAF1-dependent and -independent mechanisms to modulate DDR signaling and histone PTMs (Fig 5A; S1 Data). To further confirm these observations, we depleted *DCAF1* mRNA utilizing two previously validated shRNA constructs [13,55]. Transient expression of knockdown constructs in HeLa cells resulted in robust depletion of DCAF1 protein and mRNA 72 hours post-transfection (S3E Fig). Sequential transfection and infection experiments were performed to determine the impact of *DCAF1* depletion on cells infected with $Vpr_{WT}$ or control viruses. Knockdown of DCAF1 recapitulated our findings using $Vpr_{Q65R}$ and $Vpr_{H71R}$ viruses, further supporting the model that Vpr utilizes DCAF1-dependent and -independent mechanisms to drive DDR activation and histone PTMs (Fig 5B; S1 Data).

A recent study demonstrated that Vpr expression leads to dysregulation of histone 1 ubiquitination and that pharmacological inhibition of the CUL4/DDB1/DCAF1 complex could recapitulate this phenotype in the absence of infection [14], raising the possibility that hijacking of the CUL4/DDB1/DCAF1 complex itself may impact the epigenetic landscape. DCAF1 has been shown to preferentially interact with unmodified histone 3 N-terminal tails, which facilitates histone deacetylase recruitment to chromatin while simultaneously inhibiting histone acetyltransferase recruitment [56,57]. Consistent with this scenario, DCAF1 was relocalized from the nucleus to the cytoplasm in primary MDMs, differentiated THP1, and HeLa cells infected with $Vpr_{WT}$ virus but not in cells infected with control or $Vpr_{Q65R}$/$Vpr_{H71R}$ viruses (Fig 5C). Taken together, these observations support recent studies that suggest Vpr-induced DDR signaling is only partially dependent on DCAF1 engagement [17,58].

We next probed DCAF1-independent mechanisms by leveraging the Vpr loss-of-function mutants identified in Fig 4. N-terminally eGFP-tagged wild-type and loss-of-function Vpr proteins were generated and evaluated using live cell fluorescence microscopy to assess protein abundance and subcellular localization. As depicted in Fig 6A and 6B, all constructs exhibited whole cell distribution with nuclear enrichment, which is consistent with prior studies [59,60]. However, permeabilization of live cells resulted in a significant loss of eGFP fluorescence in cells expressing Vpr loss-of-function mutants but not in cells expressing $Vpr_{WT}$, which exhibited eGFP accumulation at the nuclear envelope and in the nucleoplasm (Representative images in Fig 6A and 6B, quantification in 6C and S1 Data).

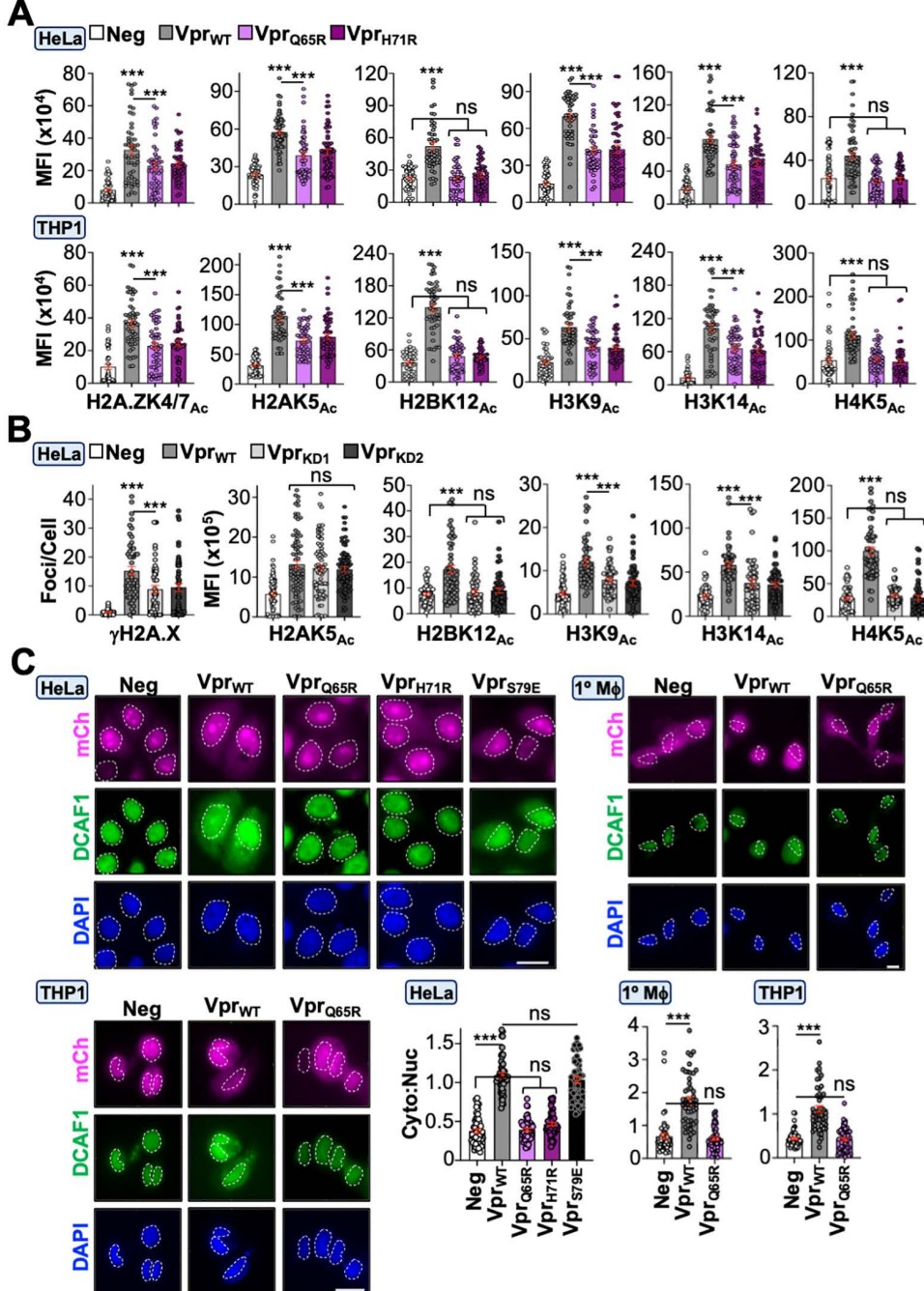

**Fig 5. Vpr utilizes DCAF1-dependent and -independent mechanisms to trigger DDR and epigenetic remodeling. (A)** Immunofluorescence microscopy quantification of histone marks in HeLa and differentiated THP1 cells infected with indicated viruses 48 hours post-infection ($n = 50$ cells). Analyses performed using a one-way ANOVA; ns, not significant; *** $p < 0.001$. The data underlying this Figure can be found in S1 Data. **(B)** Immunofluorescence microscopy quantification of histone marks following infection of HeLa cells expressing control or *DCAF1* knockdown constructs 48 hours post-infection ($n = 50$ cells). HeLa cells were sequentially transfected and infected with the indicated provirus and a shRNA construct that expresses a BFP reporter from a separate promoter. mCherry and BFP expressing cells were validated for loss of DCAF protein and changes to histone marks. Analyses performed using a one-way ANOVA; ns, not significant; *** $p < 0.001$. The data underlying this Figure can be found in S1 Data. **(C)** Representative fluorescence microscopy images and quantification of DCAF1 localization in HeLa, MDMs, and differentiated THP1 cells infected with the indicated viruses 48 hours post-infection ($n = 50$ cells). Analyses performed using a one-way ANOVA; ns, not significant; *** $p < 0.001$. The data underlying this Figure can be found in S1 Data.

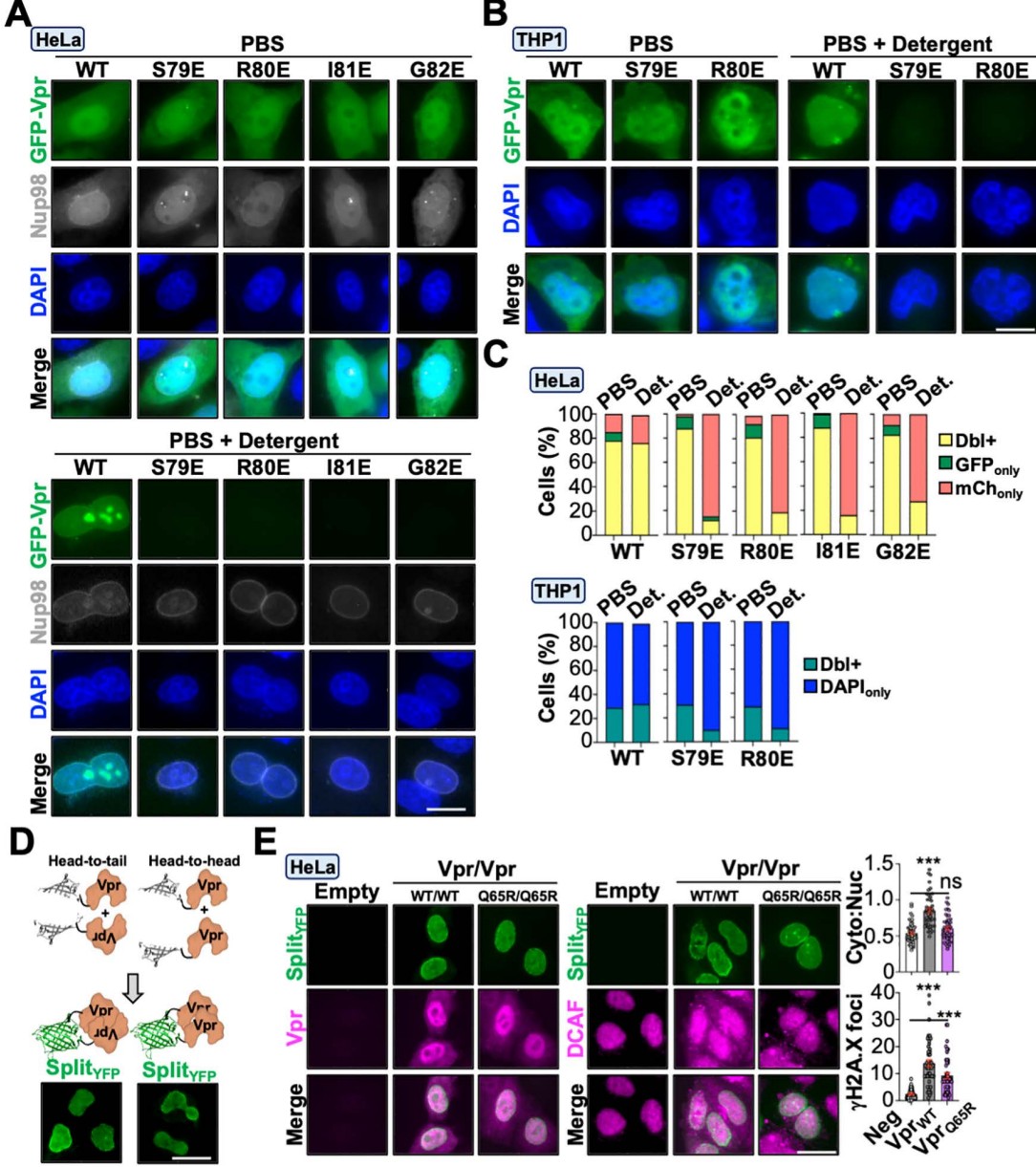

**Fig 6. Vpr stratifies into two functional pools with distinct activities. (A, B)** Representative live cell fluorescence microscopy images of HeLa or differentiated THP1 cells expressing indicated constructs 48 hours post-transfection or -infection, respectively. For washout experiments, cells were either untreated or incubated with detergent for 15 min prior to live-cell imaging. **(C)** Live-cell fluorescence quantification of dual or singly fluorescence-positive cells from Fig 6A and 6B (*n* = 50 cells). The data underlying this Figure can be found in S1 Data. **(D)** Diagrams of BiFC co-transfections and representative live cell fluorescence microscopy images of HeLa cells co-expressing indicated Vpr constructs. **(E)** Representative fluorescence microscopy images of Vpr (left) and DCAF1 (right) localization in HeLa cells infected with indicated viruses with quantification of DCAF1 cytoplasmic-to-nuclear ratio (Top) and induction of γH2A.X foci (Bottom) (*n* = 45 cells). Cells were co-transfected with NTvenus-Vpr and CTvenus-Vpr constructs for 24 hours prior to live-cell imaging. Analyses performed using a one-way ANOVA; ns, not significant; *** *p* < 0.001. The data underlying this Figure can be found in S1 Data.

Because surface mapping studies indicated that Vpr may utilize electrostatic interactions to trigger DDR signaling and histone PTMs, we reasoned that nuclear retention of Vpr was due to chromatin binding, which has been previously observed for recombinant Vpr *in vitro* [61–63]. To test this, we performed nuclear and cytoplasmic cell fractionations on HeLa cell lysates infected with $Vpr_{WT}$ or mutant viruses 48-hours post-infection. As anticipated, $Vpr_{WT}$ was predominantly observed in the nuclear fraction whereas $Vpr_{S79E}$ and $Vpr_{R80E}$ were enriched in the cytoplasmic fraction, suggesting that nucleus-localized Vpr may be binding chromatin (S3F Fig; S1 Raw Images). In addition, it has been previously suggested that Vpr's ability to bind nucleic acids *in vitro* requires oligomerization [61]. We utilized bimolecular fluorescence complementation to determine if nucleus-localized Vpr forms oligomers *in cellulo*. Co-expression of either N- or C-terminally-tagged Vpr proteins fused to split-Venus fragments exhibited robust fluorescence reconstitution at the nuclear envelope and in the nucleoplasm, but not in the cytoplasm (Fig 6D). These observations raised the possibility that Vpr exists in two functionally discrete pools, one associated with chromatin in the nucleus and the other bound to DCAF1 in the cytoplasm. To test this possibility, we leveraged the split-fluorescence system as a live-cell indicator of Vpr localization and function. First, we evaluated Vpr localization in cells that exhibited fluorescence reconstitution using a polyclonal antibody that recognizes the Vpr split-Venus fusion protein (Fig 6E, left). Immunolabeling of HeLa cells transiently expressing $Vpr_{WT}$ and $Vpr_{Q65R}$ proteins detected Vpr proteins in the nucleus and cytoplasm whereas fluorescence reconstitution only occurred in the nucleus (Fig 6E, left). Next, we verified that the Vpr split-Venus fusion protein was functional by assessing DCAF1 relocalization and DDR activation. We observed redistribution of DCAF1 to the cytoplasm and induction of γH2A.X foci in cells expressing $Vpr_{WT}$, but nucleus-localized DCAF1 and diminished γH2A.X activation in cells expressing $Vpr_{Q65R}$ (Fig 6E, right and S1 Data), suggesting that Vpr exists in two functionally discrete pools.

## Vpr-induced DDR signaling enhances HIV-1 transcription during acute infection and latency reactivation

To determine the virologic function of these Vpr activities, we generated proviruses that express mCherry from the native HIV-1 promoter (schematic in Fig 7A). Differentiated THP1, primary MDMs, or HeLa cells infected with $Vpr_{WT}$ virus exhibited significantly increased mCherry expression compared to control infected cells at all doses tested (Figs 7A, 7B, and S4A; S1 and S2 Data). Importantly, treatment with caffeine, ATM/ATR inhibitors, or infection using $Vpr_{S79E}$ reduced Vpr-enhanced mCherry expression to that of control infection (Figs 7A–7C and S4A). To confirm increased LTR activity results in increased HIV-1 protein expression, we utilized a Gag-BFP provirus. Cells expressing $Vpr_{WT}$ provirus exhibited an increase in Gag-BFP expression compared to Vpr-deficient control provirus, which could be ablated by inhibition of DDR signaling (Fig 7D; S1 Data).

Recent studies have suggested that Vpr can reactivate virus from latency, although the mechanism of action has not been fully elucidated [27–29,64,65]. Because Vpr-induced DDR activation correlates with enhanced HIV-1 promoter activity during acute infection (Fig 7A–7D), we reasoned that it would also promote latency reversal. To test this, we generated a clonal "off-to-on" HeLa latency model (H-Lat) that stably expresses the HIV 5′-LTR promoter upstream of an eGFP expression cassette, allowing for direct correlation between promoter activation and Vpr expression in the absence of *tat* or *vpr* expression from an integrated provirus. Under steady-state conditions, eGFP expression was not detectable by fluorescence microscopy or flow cytometry (Fig 7E). H-Lat cells were infected with increasing amounts of $Vpr_{WT}$ or control viruses, and eGFP expression was assessed via flow cytometry (Fig 7F; S2 Data). Infection of H-Lats with $Vpr_{WT}$ virus induced significantly higher eGFP expression compared to control infections, regardless of MOI, which could be ablated by caffeine or ATM/ATR inhibitors (Fig 7F and 7G; S1 and S2 Data). Importantly, infection of three additional latency models exhibited increased virus reactivation from latency in the presence of Vpr, which could be significantly diminished following DDR inhibition (Fig 7H; S1 and S2 Data). Interestingly, transient expression of Tat in combination with etoposide, which induces DNA double-strand breaks, induced H-Lat eGFP expression to the same magnitude as co-expression of Tat and Vpr, further supporting the role of DDR signaling in HIV-1 promoter activation (Fig 7I; S1 and S2 Data).

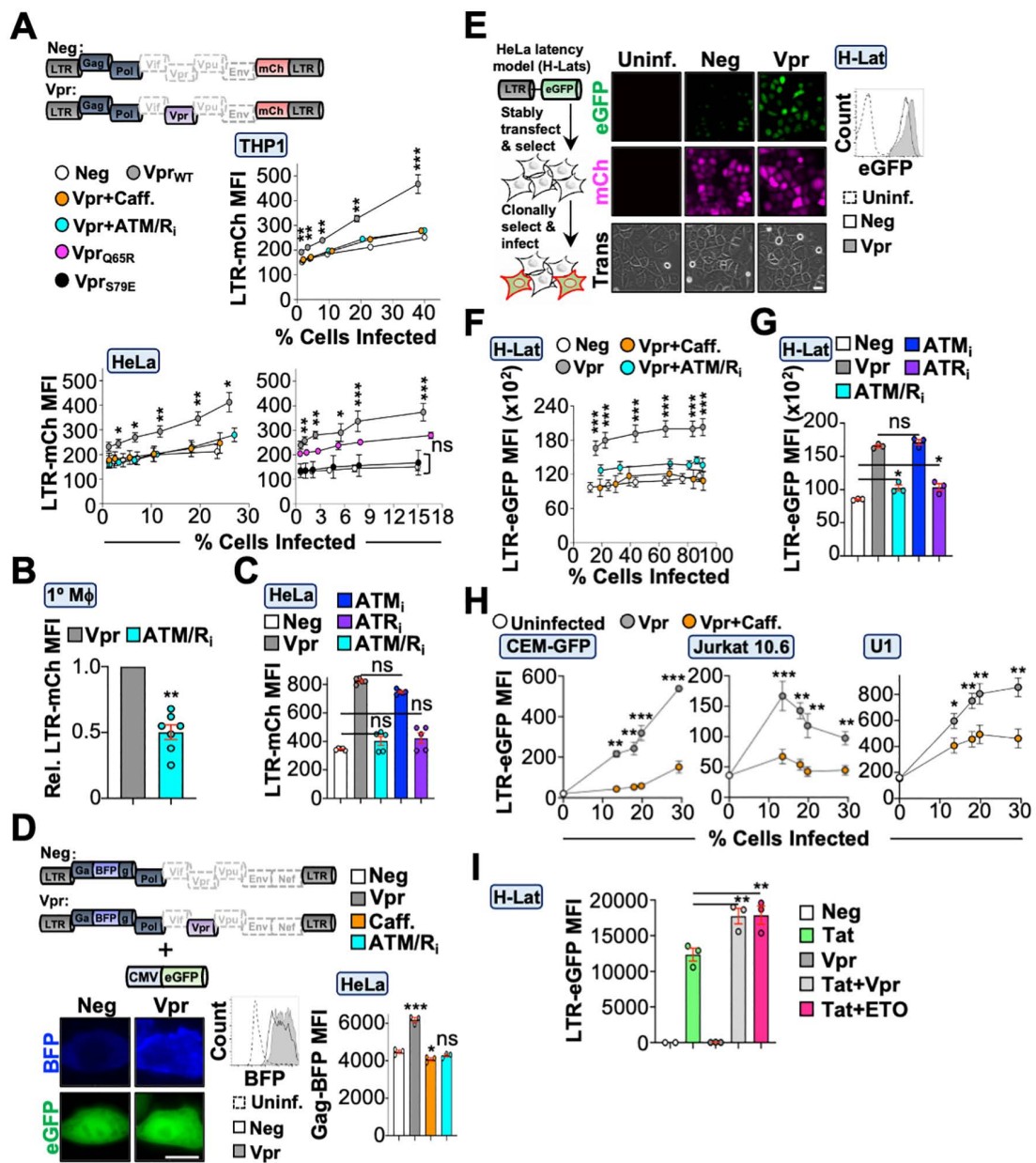

**Fig 7. Vpr induces HIV promoter activity through the DDR response. (A)** Top, schematic of proviruses used in these experiments. For these constructs, *mCherry* expression is driven off of the HIV promoter as opposed to being driven off of CMV. Bottom, flow cytometric analysis of HIV LTR activity in THP1 or HeLa cells infected with increasing MOI of the indicated viruses (*n* = 3 experiments). For inhibitor treatments, cells were infected for 24 hours prior to a 24-hour treatment with 3 mM caffeine or combined ATM (10 nM)/ATR (10 μM) treatment. ns, not significant; * *p* < 0.05; ** *p* < 0.01; *** *p* < 0.001. The data underlying this Figure can be found in S1 Data. Representative gating strategies are depicted in S2 Data and raw FSC files can be found in the Figshare Data repository (https://doi.org/10.6084/m9.figshare.c.8239897). **(B)** Flow cytometric quantification of HIV LTR activity in primary MDMs infected with Vpr$_{WT}$ virus in the presence or absence of DDR inhibition (*n* = 7 experiments). MDMs were infected for 24 hours prior to treating with combined ATM (10 nM)/ATR (10 μM) inhibitors for 24 hours. Cells were detached and mCherry fluorescence was quantified via flow cytometry. ** *p* < 0.01. The data underlying this Figure can be found in S1 Data. Representative gating strategies are depicted in S2 Data and raw FSC files can be found in the Figshare Data repository (https://doi.org/10.6084/m9.figshare.c.8239897). **(C)** Flow cytometric quantification of LTR-mCh MFI in infected HeLa cells subjected to ATM (10 nM), ATR (10 μM), or combined ATM and ATR inhibition 48 hours post-infection (n = 5 experiments). Cells were detached and mCherry fluorescence was quantified via flow cytometry. ns, not significant. The data underlying this Figure can be found in S1 Data. Representative gating strategies are depicted in S2 Data and raw FSC files can be found in the Figshare Data repository (https://doi.org/10.6084/m9.figshare.c.8239897). **(D)** Top, schematic of proviruses used in these experiments. For these proviruses, *BFP* was incorporated into the *gag* gene to generate Gag-BFP

proteins from the viral promoter. Bottom left, live cell fluorescence microscopy images of HeLa cells 48-hours post-co-transfection with the indicated Gag-BFP proviruses and an eGFP control plasmid. Bottom middle, flow cytometric histogram of BFP fluorescence in eGFP-positive HeLa cells. Bottom right, quantification of BFP MFI in eGFP-positive cells with or without DDR inhibition ($n = 3$ experiments). Cells were co-transfected for 24 hours prior to adding the indicated inhibitor for 24 hours. Analyses performed using a one-way ANOVA; ns, not significant; *** $p < 0.001$; * $p < 0.05$. The data underlying this Figure can be found in S1 Data. Representative gating strategies are depicted in S2 Data and raw FSC files can be found in the Figshare Data repository (https://doi.org/10.6084/m9.figshare.c.8239897). **(E)** Diagram displaying the establishment of the HeLa latency model H-Lats (left) and representative live cell fluorescence microscopy images of H-Lat GFP expression when infected with indicated viruses. H-Lat cells were infected for 48 hours prior to being subjected to live cell fluorescence microscopy. Right, flow cytometric analysis of H-Lat GFP expression 48 hours post-infection with the indicated viruses. **(F)** Flow cytometric analysis of H-Lat reactivation following infection with increasing MOI of the indicated viruses in the presence or absence of DDR inhibition ($n = 3$ experiments). Cells were infected for 24 hours prior to inhibitor treatment for 24 hours and preparation for flow cytometry. *** $p < 0.001$. The data underlying this Figure can be found in S1 Data. Representative gating strategies are depicted in S2 Data and raw FSC files can be found in the Figshare Data repository (https://doi.org/10.6084/m9.figshare.c.8239897). **(G)** Flow cytometric quantification of LTR-eGFP MFI in infected H-Lat cells subjected to ATM (10nM), ATR (10 µM), or combined ATM/ATR treatment. H-Lat cells were infected for 24 hours prior to a 24-hour inhibitor treatment and preparation for flow cytometry ($n = 5$ experiments). ns, not significant; * $p < 0.05$. The data underlying this Figure can be found in S1 Data. Representative gating strategies are depicted in S2 Data and raw FSC files can be found in the Figshare Data repository (https://doi.org/10.6084/m9.figshare.c.8239897). **(H)** Flow cytometric analysis of CEM-GFP, J-Lat clone 10.6, and monocyte clone U1 latency models uninfected or infected with Vpr$_{WT}$ virus in the presence or absence of 3 mM caffeine ($n = 3$ experiments). Cells were infected with increasing MOI of Vpr-expressing virus for 24 hours, treated with vehicle or 3 mM caffeine for 24 hours, and then subjected to flow cytometry. * $p < 0.05$; ** $p < 0.01$; *** $p < 0.001$. The data underlying this Figure can be found in S1 Data. Representative gating strategies are depicted in S2 Data and raw FSC files can be found in the Figshare Data repository (https://doi.org/10.6084/m9.figshare.c.8239897). **(I)** Flow cytometric quantification of LTR-eGFP MFI in H-Lat cells transiently expressing plasmids encoding Tat and/or Vpr and treated with 50 µM etoposide or vehicle ($n = 3$ experiments). H-Lats were transfected with the indicated plasmid combination for 24 hours, treated with vehicle or etoposide for 24 hours, and then subjected to flow cytometry. Analyses performed using a one-way ANOVA; ** $p < 0.01$. The data underlying this Figure can be found in S1 Data. Representative gating strategies are depicted in S2 Data and raw FSC files can be found in the Figshare Data repository (https://doi.org/10.6084/m9.figshare.c.8239897).

## Vpr-directed DDR signaling activates transcription programs and transcription-coupled R-loops to enhance HIV-1 promoter activity

We sought to further define the molecular mechanism(s) underlying enhanced HIV-1 promoter activity. Recent studies have indicated that during acute HIV-1 infection DNA-PK can directly phosphorylate RNA polymerase II (RNA$_{Pol\ II}$) to promote HIV-1 transcription initiation and elongation [30,31]. We reasoned that Vpr's ability to enhance HIV-1 promoter activity may also utilize this mechanism as we demonstrate that DDR signaling is a prerequisite (Fig 7). First, we evaluated Vpr's ability to induce DNA-PK activation. Virion-associated and *de novo* Vpr proteins induced a significant increase in DNA-PK phosphorylation that could be ablated by treatment with two different inhibitors targeting DNA-PK autophosphorylation (NU7926 and NU7441; S4B and S4C Fig; S1 Data). Interestingly, concomitant inhibition of ATM and ATR also ablated DNA-PK phosphorylation, with individual inhibitor treatments indicating that ATR activation is required for DNA-PK activation (S4B Fig; S1 Data). Moreover, while DNA-PK activity is not required for Vpr-induced DDR signaling, it is required for enhanced HIV-1 promoter activity (S4D and S4E Fig; S1 and S2 Data).

Next, we assessed the activation of transcription factors previously associated with DDR signaling and are known to influence HIV-1 promoter activity. In response to DNA damage, NFκB [66,67], SP1 [68,69], and cJun [70,71] are directly phosphorylated by DNA repair kinases. Importantly, Vpr has been implicated in activating the ATM/NEMO complex [18], Vpr and Tat interact with SP1 to promote HIV-1 LTR transcription [68,72], and cJun has been shown to interact with Tat and Vpr [73,74]. Differentiated THP1 and HeLa cells were infected with Vpr-proficient or -deficient viruses and subjected to immunoblot analysis and immunofluorescence microscopy to assess the activation of NFκB, SP1, cJun, and RNA$_{Pol\ II}$. Vpr induced the phosphorylation of amino acid residues correlated with transcription factor function and at RNA$_{Pol\ II}$ residues required for transcription initiation and elongation (Figs 8A and S4F; S1 Data). Moreover, increased phosphorylation was not due to an increase in transcription factor or RNA$_{Pol\ II}$ abundance (S4F and S4G Fig; S1 Data). As anticipated, inhibition of ATM, ATR, or DNA-PK significantly reduced Vpr-induced phosphorylation of target transcription factors and RNA$_{Pol\ II}$ (Figs 8A, S4D, S4F, and S4H; S1 Data; S1 Raw Images).

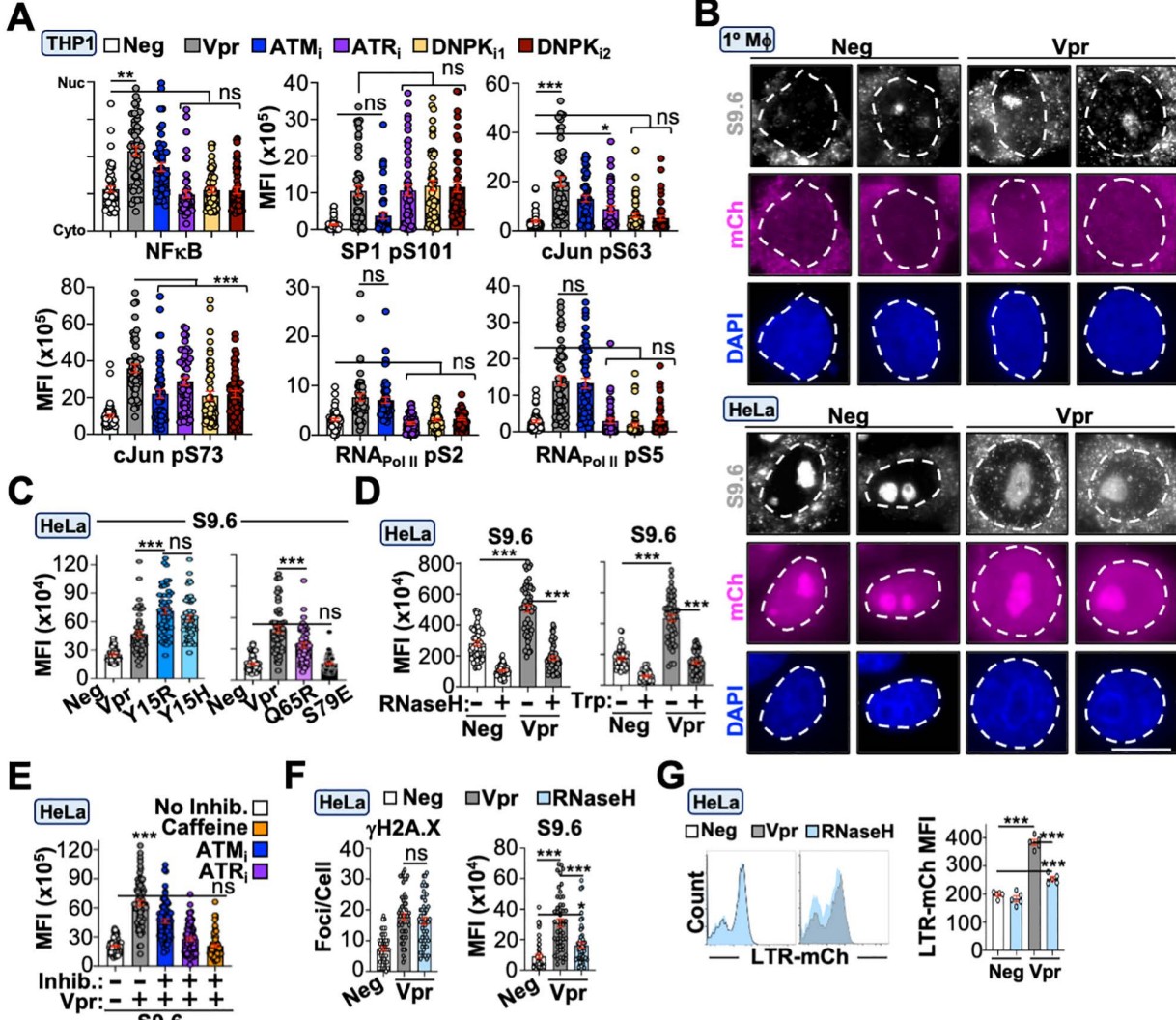

**Fig 8. Vpr-directed DDR activates transcription factors and induces transcription-coupled R-loops to promote HIV-1 transcription. (A)** Immuno-fluorescence microscopy quantification of NFκB translocation, phosphorylated SP1 residue Ser101, phosphorylated cJun residues Ser63 and Ser73, and phosphorylated RNA polymerase II residues Ser2 and Ser5 in differentiated THP1 cells infected with Vpr$_{WT}$ or control viruses in the presence or absence of ATM, ATR, or DNA-PK inhibition ($n = 50$ cells). Cells were infected for 24 hours, treated with vehicle or the indicated inhibitor for 24 hours, and then subjected to immunofluorescence microscopy. Analyses were performed using a one-way ANOVA; ns, not significant; ** $p < 0.01$; *** $p < 0.001$. The data underlying this Figure can be found in S1 Data. **(B)** Representative immunofluorescence microscopy images of R-loop abundance in primary MDM and HeLa cells infected with indicated viruses 48 hours post-infection ($n = 50$ cells). Analyses performed using a student $t$ test; *** $p < 0.001$. **(C–E)** Immunofluorescence microscopy quantification of R-loop abundance in HeLa cells infected with indicated viruses 48 hours post-infection. Samples were left untreated (C) or treated with RNaseH (D, left), triptolide (D, right), or with the indicated DDR inhibitors (E), respectively ($n = 50$ cells). For RNaseH treatment, cells were infected for 48 hours prior to methanol fixation and addition of recombinant RNaseH to deplete RNA associated with R-loops. For inhibitor treatments, HeLa cells were infected for 24 hours, treated with the indicated inhibitor for 24 hours, and then subjected to immunofluorescence microscopy. Analyses were performed using a one-way ANOVA; ns, not significant; ** $p < 0.01$; *** $p < 0.001$. The data underlying this Figure can be found in S1 Data. **(F)** Immunofluorescence microscopy quantification of DDR activation (left) and R-loop abundance (right) in HeLa cells infected with indicated viruses and transfected with RNaseH constructs ($n = 50$ cells). Cells were infected for 24 hours before transfection of control or RNaseH-expressing plasmids for 24 hours prior to being prepared for immunofluorescence microscopy. Analyses performed using a one-way ANOVA; ns, not significant; *** $p < 0.001$; * $p < 0.05$. The data underlying this Figure can be found in S1 Data. **(G)** Flow cytometric histograms of HeLa cells infected with control or Vpr$_{WT}$ LTR-mCh viruses for 24 hours prior to transient expression of RNaseH for 24 hours and quantification of LTR-mCh MFI in transfected cells via flow cytometry ($n = 3$ experiments). Analyses performed using a one-way ANOVA; ns, not significant; *** $p < 0.001$. The data underlying this Figure can be found in S1 Data. Representative gating strategies are depicted in S2 Data and raw FSC files can be found in the Figshare Data repository (https://doi.org/10.6084/m9.figshare.c.8239897).

A strong correlation has been established between the activation of DDR responses and R-loop accumulation at sites of DNA damage [75–77]. R-loops occur when nascent RNA re-anneals to the transcribed DNA strand, creating a 3-stranded structure containing an RNA/DNA hybrid and an unpaired non-transcribed DNA strand. R-loops generally accumulate at highly transcribed DNA regions and can impede replication forks, potentially leading to DNA strand breaks [78]. We utilized a well-characterized monoclonal antibody that binds RNA/DNA hybrids with high affinity to determine if Vpr expression correlates with changes in R-loop abundance [79]. Primary MDMs and HeLa cells infected with $Vpr_{WT}$ or $Vpr_{Y15R}$/$Vpr_{Y15H}$ viruses exhibited a significant increase in R-loop abundance compared to control infected cells 48-hours post-infection (representative images in Fig 8B, quantification in Figs 8C and S4I; S1 Data). We also investigated the requirement for DCAF1-engagement and DDR signaling/epigenetic remodeling activity in R-loop accumulation. $Vpr_{Q65R}$-infected cells exhibited an ~50% reduction in R-loop abundance whereas cells infected with $Vpr_{S79E}$ virus exhibited levels comparable to control infected cells, indicating that R-loop formation occurs through both DCAF1-dependent and -independent mechanisms (Fig 8C; S1 Data). To confirm these were *bona-fide* R-loops, infected cells were treated with RNase H which specifically degrades the RNA component of RNA/DNA hybrids [79,80]. Both control and $Vpr_{WT}$ infected cells exhibited a significant reduction in immunolabeling compared to vehicle-treated controls, suggesting these were authentic R-loops (Fig 8D, left and S1 Data). We also confirmed R-loops were associated with transcription by treating infected cells with the transcriptional inhibitor triptolide [81]. As depicted in Fig 8D (right), triptolide treatment significantly reduced R-loop abundance in both control and $Vpr_{WT}$-infected cells, suggesting these were occurring co-transcriptionally. Furthermore, inhibition of DDR signaling through either caffeine treatment or ATM/ATR-specific inhibitors also ablated Vpr-induced R-loops, suggesting these phenotypes are tightly associated (Fig 8E; S1 Data). To determine if Vpr-induced R-loops influence HIV-1 promoter activity, we transiently expressed RNaseH in live cells infected with Vpr and control viruses and assessed promoter activity. Overexpression of RNaseH1 resulted in significantly reduced S9.6 accumulation and HIV-1 promoter activity in Vpr-infected cells but not control-infected cells (Fig 8F and 8G; S1 and S2 Data). Taken together, these observations support a model wherein Vpr-induced DNA damage triggers parallel responses that activate transcription programs and likely chromatin opening to enhance HIV-1 promoter activity during acute infection and reactivation from latency.

## Discussion

While Vpr has been studied extensively, direct cause-and-effect mechanism(s) underlying its proviral function are unclear. Here, we demonstrate that Vpr's ability to induce DDR signaling promotes epigenetic remodeling and enhances HIV-1 promoter activity during acute infection and virus reactivation from latency. Vpr mutants that are hyperactive or deficient for DDR activation exhibit corresponding phenotypes for epigenetic remodeling and modulating HIV-1 promoter activity, suggesting a direct cause-and-effect relationship. Vpr determinants required for these activities were mapped to distinct surface regions that are electrostatic in nature and likely promote chromatin association to induce nucleus remodeling. Biochemical, genetic, and pharmacologic experiments indicate that Vpr utilizes both degradation-dependent and -independent mechanisms to induce epigenetic remodeling and likely exists as two discrete functional pools. Importantly, functional studies and phylogenetic analyses combine to indicate these Vpr activities are common in HIV-1 subtypes circulating globally. Lastly, we demonstrate that virion-associated and *de novo* Vpr induce DDR signaling and activation of transcriptional machinery to promote virus transcription.

Our findings here support a model wherein Vpr-induced DNA damage triggers parallel responses that activate transcription factors and chromatin opening to enhance HIV-1 promoter activity during acute infection and reactivation from latency. We demonstrate that Vpr induces DDR-dependent acetylation of several histone residues known to be dynamically modulated during DDR [32–36,82] and that have been linked to transcriptionally active euchromatin [32,83] (Fig 1). These observations are consistent with reports demonstrating that histone acetyltransferases p300/CBP and GCN5 promote HIV-1 transcription [39,84,85], and that pan-HDAC (histone deacetylase) inhibitors reactivate latent reservoirs

[42,43,86–88]. Because current LRAs are inefficient at reactivating diverse viral reservoirs in vivo, future studies that carefully define epigenetic synergy between Vpr and HDAC inhibitors may inform the development of novel LRAs that successfully reactivate a larger proportion of latent reservoirs.

Moreover, we establish that Vpr expression correlates with the activation of several transcription factors known to influence HIV-1 promoter activity and increased R-loop abundance (Fig 8). Interestingly, two recent studies suggest HIV-1 integration sites are enriched at R-loop-containing hotspots [89,90]. Given that we demonstrate virion-associated Vpr induces DDR signaling prior to provirus integration (Fig 3), it is possible that this activity enhances integration efficiency through the generation of R-loops. Additional experimentation is necessary for testing this possibility and would further define the functional role of virion-associated Vpr. Importantly, our functional and bioinformatic analyses of common HIV-1 Group M subtypes and clinical isolates indicate that induction of DDR signaling and epigenetic remodeling activities are broadly conserved (Figs 2 and 4). Thus, future studies aimed at developing small molecule compounds that block these Vpr functions would likely exhibit broadly-spectrum efficacy against diverse viral strains circulating globally.

While Vpr is thought to primarily function through the depletion of host factors, emerging evidence supports a model wherein Vpr also functions independent of E3-ligase engagement. For instance, structural studies have demonstrated that Vpr binds the nucleotide-excision repair protein RAD23A using the same interface as DCAF1 and that binding is mutually exclusive [91]. Additionally, Vpr mutants that fail to bind DCAF1 or deplete canonical DNA repair substrates can still induce DNA strand breaks, activate DDR signaling, and exert a proviral effect during acute infection [17,18,92]. These observations raise the possibility that Vpr exists in at least two functional pools that have distinct activities. In support of this model, several previous studies have indicated that Vpr can engage nucleic acids and chromatin in vitro and *in cellulo* to activate ATR signaling, alter chromatin structure and segregation, and trigger cell cycle arrest [16,61,93–95]. Moreover, our findings further support this model by indicating Vpr-induced epigenetic remodeling occurs through DCAF1-dependent and independent mechanisms, and that nucleus-localized Vpr is multimeric while cytoplasmic Vpr likely exists as soluble monomers (Fig 6). Interestingly, several studies have demonstrated that Vpr is phosphorylated at residues S79, S94, and S96 [96–98] and can engage 14-3-3 proteins, which specifically bind phosphorylated targets [99,100]. These observations raise the possibility that phosphorylation of the C-terminal tail may regulate discrete Vpr functions. In support of this idea, Vpr S79A has been shown to induce similar levels of γH2A.X foci compared to wild-type Vpr but loses the ability to induce G2/M cell cycle arrest [17,96]. Taken together, our observations support the notion that these Vpr activities are advantageous for virus replication and provide mechanistic rationale for its function in virus infection establishment.

## Experimental model and subject details

**Experimental replicates and statistical analyses.** All experimental procedures were repeated at least three independent times. MDMs were derived using PBMCs from four independent donors. All flow cytometry data were analyzed using FlowJo v10 software. Focus formation and MFI analyses were calculated using ImageJ software and analyzed using GraphPad Prism 6 software. Briefly, boundaries of infected cell nuclei were defined using DAPI staining as an indicator and then foci were quantified using the "find maxima" feature and eGFP MFI was defined by analyzing integrated pixel intensity of the defined nuclear area minus the background signal intensity of an adjacent area with identical dimensions. Foci prominence thresholding was set so that uninfected negative controls yielded minimal foci. The Vpr-DCAF co-structure depicted in Fig 3 was generated using Chimera protein modeling software (PDB: 5JK7). The schematic in Fig 1C was generated using BioRender (z50b224). Statistical analyses were performed using either an unpaired two-tailed Student *t* test, a one-way ANOVA, or a two-way ANOVA using GraphPad Prism 8 after confirming that all data followed a normal distribution.

**Cell lines and culture conditions.** HeLa and HEK293T cells (American Type Culture Collection) were maintained in DMEM medium (Gibco cat #11-965-118) supplemented with 10% fetal bovine serum (FBS; Gibco, Gaithersburg, MD) and 0.5% penicillin-streptomycin (50 units; Gibco, Gaithersburg, MD). HeLa/HEK293T cells were transfected using 1 mg/ml

polyethylenimine (PEI; Fisher #NC1014320) at a ratio of 3 μL per 1 μg of DNA. For virus generation, HEK293T cells were co-transfected with a VSV-G expression vector along with the indicated proviral plasmid. For virus generation in *trans*-complementation studies, HEK293T cells were co-transfected with a VSV-G expression vector, the mCh control provirus, and a pcDNA expression vector encoding Vpr$_{WT}$ or Vpr$_{S79E}$. Medium was collected 48 h post-transfection and frozen at −80 °C until used. THP1 cells were maintained in RPMI medium (Gibco; cat #11-875-093) supplemented with 10% FBS and 0.5% penicillin-streptomycin. For THP1 differentiation, cells were incubated with 50–100 ng/ml of phorbol 12-myristate 13-acetate (PMA; Sigma #P8139) for 48–96 h. Primary MDMs were isolated via plastic adhesion of human peripheral blood mononuclear cells (hPBMCs; Lonza cat #CC-2704). To stimulate differentiation, cells were incubated with 50 ng/ml of granulocyte-macrophage colony-stimulating factor (GM-CSF; PeproTech cat #300-03) for 72–96 h until macrophage morphology was observed. For generating the HIV LTR-GFP cell line, HeLa cells were stably transfected with a previously described 5′-LTR eGFP expression vector for 48 h before selecting with 1 mg/ml puromycin (Thermo Fisher, BP295100) to generate pure cell populations. Clonal isolates were generated using limiting dilution in 96-well flat bottom culture plates and validated for reactivation efficiency using HDAC$_i$ treatment.

## Method details

**Plasmids and cloning.** Proviral expression plasmids have been described previously [101]. Briefly, for both the control and Vpr-expressing viruses, two amino acid mutations (I31A, R33A) were introduced in Vif to inhibit interactions with a host phosphatase complex that has a role in DNA repair [102]. For the control virus, Vpr has two stop codons introduced at amino acid positions 23 and 24. Lastly, *vpu* has been deleted and the 5′ region of *eEnv* has been deleted to inhibit virus spread while keeping the rev-response element intact. For *DCAF1* knockdown, previously validated shRNA sequences were cloned into a pLKO plasmid expressing mTag-BFP2 in place of puromycin (KD$_1$, GCTGAGAATACTCTTCAAGAA; KD$_2$, TCACAGAGTATCTTAGAGA) [13,48]. Vpr proteins or eGFP-tagged Vpr proteins and Nup98-mCherry were cloned into a pcDNA 5TO expression vector. eGFP-tagged Vpr proteins were also clones into a pLenti expression vector. Vpr single amino acid substitution mutants were generated by PCR amplification using Phusion high fidelity DNA polymerase (NEB, Ipswich, MA) and overlapping PCR. The RNAseH1 expression plasmid was obtained through Addgene and the mCherry cassette was exchanged with mTag-BFP2 (pICE-RNaseH1-WT-NLS-mCherry; #60365). All constructs were confirmed by Sanger sequencing and restriction enzyme digestions.

**Fluorescence microscopy and immunostaining.** For immunofluorescence microscopy experiments using HeLa cells, approximately 5,000 cells were seeded into a 96-well glass-bottom imaging plate (Ibidi #89627) and allowed to adhere overnight at 37 °C. The next day, cells were infected with the indicated virus for 48 hours prior to being washed 1× with phosphate-buffered saline (PBS) and then fixed using 4% paraformaldehyde (PFA) for 10 min at room temperature. After fixation, cells were washed 3× with PBS in 5-min intervals and permeabilized using PBS plus 0.3% Triton X-100 (PBST) for 10 min at room temperature. Cells were blocked using PBST supplemented with 5% bovine serum albumin (BSA, Fisher Bioreagents BP9703100), 10% goat serum (Sigma-Aldrich, G9023-10mL), and 0.3 M glycine for 2 hours at room temperature while rocking. After blocking, samples were incubated with primary antibodies against γH2A.X (1:300, Cell Signaling 9718), pCHK1 (1:50, Cell Signaling 2348), pCHK2 (1:200, Cell Signaling 2661), acetyl-histone H2AK5 (1:1,000, Cell Signaling 2576), acetyl-histone H2A.Z K4/K7 (1:1,000, Cell Signaling 75336), total H2B (1:1,000, Cell Signaling 12364), acetyl-histone H2BK5 (1:1,000, Cell Signaling 12799), acetyl-histone H2BK12 (1:1,000, Cell Signaling 5410), acetyl-histone H2BK15 (1:1,000, Cell Signaling 9083), acetyl-histone H2BK20 (1:1,000, Cell Signaling 34156), total H3 (1:1,000, Cell Signaling 4499), phospho-histone H3S10 (1:1,000, Cell Signaling 53348), acetyl-histone H3K9 (1:1,000, Cell Signaling 9649), tri-methyl-histone H3K9 (1:1,000, Cell Signaling 13969), acetyl-histone H3K14 (1:1,000, Cell Signaling 7627), acetyl-histone H3K18 (1:1,000, Cell Signaling 13998), acetyl-histone H3K27 (1:1,000, Cell Signaling 8173T), tri-methyl histone H3K27 (1:1,000, Cell Signaling 9733), total H4 (1:1,000, Cell Signaling 2935), acetyl-histone H4K5 (1:1,000, Cell Signaling 8647), acetyl-histone H4K8 (1:1,000, Cell Signaling 2594), acetyl-histone

H4K12 (1:1,000, Cell Signaling 13944T), Rpb1 CTD (1:1,000, Cell Signaling 2629T), Phospho-Rpb1 (Ser2) (1:1,000, Cell Signaling 13499T), Phospho-Rpb1 (Ser5) (1:1,000, Cell Signaling 13523T), DNA-PKcs (1:200, Cell Signaling 12311S), DNA-PKcs (phospho S2056) (1:200, Abcam ab18192), DCAF1/VprBP (1:400, Proteintech 11612-1-AP), NF-κB p65 (1:300, Cell Signaling 82425S), c-Jun (1:400, Cell Signaling 9165T), Phospho-c-Jun (Ser63) (1:1,000, Cell Signaling 91952T), Phospho-c-Jun (Ser73) (1:800, Cell Signaling 3270T), Sp1 (1:500, Cell Signaling 9389T), Phospho-Sp1 (Ser101) (1:300, Active Motif 39758), or S9.6 DNA-RNA hybrid (1:200, Kerafast ENH001) in blocking buffer overnight at 4 °C. The next day, cells were washed 3× with PBS in 5-min intervals and then incubated with anti-mCherry conjugated to Alexa Fluor 594 (1:800, Invitrogen M11240), secondary anti-rabbit-IgG conjugated to Alexa Fluor 488 (1:800, Cell Signaling 4412), secondary anti-rabbit-IgG conjugated to Alexa Fluor 594 (1:800, Cell Signaling 8889) or secondary anti-mouse-IgG conjugated to Alexa Fluor 488 (1:800, Cell Signaling 4408) in blocking buffer at room temperature for 1 hour. After incubation, cells were washed 3× with PBS at 5-min intervals and stained with NucBlue stain (Thermo Fisher, R37605) and imaged. An EVOS M500 fluorescence microscope was used for imaging, using 60× or 100× oil-immersion objectives. For immunofluorescence microscopy experiments using differentiated THP1 cells, approximately 20,000 cells were seeded into a 96-well glass-bottom imaging plate in the presence of 50–70 ng/ml PMA and co-infected with the indicated virus for 48–96 h prior to being subjected to immunofluorescence microscopy. For immunofluorescence experiments using primary MDMs, approximately 15,000 macrophages were seeded into a 96-well glass-bottom imaging plate 5 days post-differentiation with GM-CSF. The next day, cells were infected for 48 hours prior to being prepared for immunofluorescence microscopy.

**Inhibitor treatments.** For inhibitor experiments, cells were treated 24-hours post-infection with either 10 nM ATM inhibitor (Fisher, #AZD1390), 10 μM ATR inhibitor (Fisher, #NU6027), 3 mM caffeine, 16 μM DNA-PK inhibitor #1 (MedChemExpress, #NU7026), or 16 μM DNA-PK inhibitor #2 (MedChemExpress #NU7441) for 24 hours. Triptolide was used at 1 mM for 4 hours. RNase H-treated cells were incubated with 150 U/mL of enzyme, or buffer alone, for 2 h at 37° (NEB, M0297). Cells were then processed for downstream analysis as indicated. For experiments using ART, cells were pretreated with 5 μM raltegravir (NIH HIV Reagent Program, HRP-11680) or with a combination of 10 μM zidovudine (AZT) (NIH HIV Reagent Program, HRP-3485) and 100 nM etravirine (NIH HIV Reagent Program, HRP-11609), for 1 hour after which cells were infected and subjected to immunofluorescence or flow cytometry 24 hours post-infection.

**Live cell permeabilization experiments.** Approximately 5,000 HeLa cells were seeded into a 96-well glass-bottom imaging plate and allowed to adhere overnight at 37 °C. The next day, cells were co-transfected with approximately 150 ng of pcDNA eGFP-Vpr wild-type or mutant constructs and 150 ng of pcDNA Nup98-mCherry. Forty-eight h post-transfection, live cells were treated with 0.5% Triton X-100 in PBS for 5 min and imaged using an EVOS M5000 microscope. For experiments in THP1 cells, 40,000 cells were seeded, differentiated, and infected with pLenti eGFP-Vpr wild-type or mutant viruses into a 96-well glass-bottom imaging plate and treated with 50 μg/mL Digitonin prior to fixing and imaging.

**Immunoblotting.** For immunoblotting experiments, HeLa or THP1 cells were plated at a seeding density of 350,000 cells per well in a 6-well tissue culture plate and infected with the indicated virus for 48 h prior to harvesting. Cell pellets were resuspended in RIPA buffer (50 mM Tris [pH 8.0], 150 mM NaCl, 1 mM β-mercaptoethanol, 1% Triton X-100, 0.1% sodium dodecyl sulfate (SDS), 0.5% deoxycholate) supplemented with a protease and phosphatase inhibitor cocktail (Thermo Scientific #78440). Cell lysate was combined with 5× loading dye (62.5 mM Tris [pH 6.8], 20% glycerol, 5% β-mercaptoethanol, 2% SDS, 0.05% bromophenol blue) and samples were separated using a 12% SDS-PAGE gel and transferred using a 0.2 μM polyvinylidene fluoride membrane (Thermo Scientific #78440). Membranes were blocked in 5% BSA in PBST for 1 h and then incubated with primary antibodies described above, and antibodies against Vpr (1:500, Proteintech 51143-1-AP) and tubulin (1:1,000, Cell Signaling 3873) diluted in blocking buffer overnight at 4 °C while rocking. The next day, primary antibody was removed, and membranes were washed 3× with PBST for 5 min each. After a brief 30-min blocking step with 5% milk in PBST, membranes were incubated with α-mouse HRP (1:10,000, SantaCruz sc-525409) or α-rabbit HRP (1:10,000, Cell Signaling 7074P2) secondary antibody diluted in 5% milk PBST and rocked for

1 h at room temperature. After washes with PBST, blots were incubated with West Pico PLUS chemiluminescent substrate (Thermo Scientific #34580) for 5 min before imaging on a BioRad ChemiDoc MP Imaging System.

**Bimolecular fluorescence complementation.**  Vpr was cloned into a pcDNA expression vector expressing CMV-driven N- or C-terminal fragments of Venus. This expression vector also encoded SV40-driven mCherry in place of hygromycin as a transfection reporter. Approximately 5,000 HeLa cells were plated into a 96-well glass-bottom plate and allowed to adhere overnight. The next day, cells were transfected with 150 ng each of N- or C-terminally tagged Vpr derivatives and imaged 24-hours post-transfection. For coupling immunofluorescence microscopy with bimolecular fluorescence complementation, 48-hours post-transfection HeLa cells were fixed with 4% PFA, and immunofluorescence labeling was performed as described above (DCAF1, 1:400, Proteintech 11612-1-AP; GFP, 1:500, Cell Signaling 2956).

**Cell fractionation.**  For fractionation experiments, SimpleChIP Enzymatic Cell Lysis Buffers A and B (Cell Signaling 14282) were utilized. Briefly, cells were infected as described above and trypsinized, washed, and fixed in 1% paraformaldehyde for 30 min. After washing in cold PBS, cells were centrifuged and resuspended in cold Buffer A supplemented with 1M dithiothreitol (DTT) and 200× protease inhibitor cocktail (PIC; Cell Signaling 7012) and incubated on ice for 10 min, with mixing by inversion every 3 min. Extracted intact nuclei were pelleted by cold centrifugation at 2,000 RCF for 5 min, the cytoplasmic fraction was removed and transferred to a fresh Eppendorf tube. Intact nuclei were washed with Buffer A and resuspended in ice-cold Buffer B supplemented with 1M DTT, pelleted at 2,000$g$ for 5 min. This process was repeated twice. To fragment genomic DNA, nuclei were treated with micrococcal nuclease (MNase; Cell Signaling 10011) and incubated for 30 min at 37 °C with frequent mixing after which this reaction was stopped using 0.2% SDS and 10 mM ethylenediaminetetraacetic acid (EDTA) stop buffer for 10 min at 37 °C. The nuclear fraction was then clarified by cold centrifugation at 9,400$g$ for 10 min and both the cytoplasmic and nuclear fractions were subjected to immunoblotting.

**Flow cytometry.**  For immunolabeling to assess histone abundance, the indicated cell lines were seeded into 12-well plates and adhered overnight prior to being infected with the indicated virus. At 48 hours post-infection, cells were washed and harvested following trypsinization, or collected in suspension for non-differentiated THP1 cells, and centrifuged at 500 RCF for 10 min. After washing with PBS, cells were centrifuged and resuspended in 4% PFA for 30 min, washed 3× with PBS, and resuspended in PBST and incubated at room temperature for 30 min. Cells were then resuspended in 5% BSA/PBST blocking buffer, incubated for 1 hour, pelleted via centrifugation at 500 RCF, and resuspended in primary antibody directed against the indicated histone mark. Samples were washed 3× times with PBS and resuspended in secondary anti-mCherry Alexa Fluor 594 and secondary anti-rabbit-IgG Alexa Fluor 488 in blocking buffer at room temperature for 1 h. Cells were washed with PBS, resuspended in cold PBS and subjected to flow cytometry using an Invitrogen Attune NxT flow cytometer. For analysis of HIV-1 LTR activity, HeLa, THP1, or primary MDMs were seeded as described above and infected with the indicated virus for 24 hours. The next day, cells were treated with vehicle or the indicated inhibitor for 24 hours. At 48 hours post-infection, cells were washed once with PBS, detached using Trypsin/EDTA or Accutase, and centrifuged at 500 RCF for 10 min at 4°. Cells were washed 1× in PBS and subjected to flow cytometry on a Becton Dickinson LSR II flow cytometer. For quantifying promoter activation from quiescence, 100,000 H-Lat cells, 100,000 J-Lat 10.6 cells (ARP-9849), or 150,000 CEM-GFP cells were seeded into a 12-well plate and infected with control virus and/or VprWT virus for 24 hours prior to caffeine treatment. At 48 hours post-infection, cells were collected and prepared for flow cytometric analysis on a Becton Dickinson LSR II flow cytometer. For quantifying promoter activation in latently infected U1 cells, 75,000 U1 cells were infected with Vpr$_{WT}$ virus for 24 hours prior to caffeine treatment. At 48 hours post-infection, 750 µL of infected supernatant was taken and used to infect 150,000 CEM-GFP cells for 48 hours, after which cells were collected and prepared for flow cytometric analysis as described above. To assess the impact of R-loops on HIV transcription, 100,000 HeLa cells were seeded into a 12-well plate and infected 24 hours later using LTR virus as indicated above. Twenty-four hours post-infection, cells were transfected for 24 hours with either an empty control plasmid or a bacterial RNaseH1-expressing plasmid as described above. Cells were then harvested and subjected to flow cytometric analysis.

**Cell cycle analysis.** For analysis of cell cycle phases, approximately 50,000 HeLa cells or primary MDMs were seeded into a 12-well plate and infected with the indicated virus. At 48 hours post-infection, cells were detached and centrifuged at 500$g$ for 10 min, resuspended in 75 mL ice-cold PBS, and fixed in 300 mL ice-cold 70% ethanol for 30 min. Cells were pelleted at 5,000$g$ for 10 min, washed twice with ice-cold PBS, and subjected to intracellular flow cytometry staining as described above. After incubation with the indicated primary and secondary antibodies, cells were resuspended in FxCycle PI/RNase staining solution and incubated per the manufacturer's protocol (Invitrogen, F10797).

**RNA extraction, cDNA synthesis, and RT-PCR.** For RT-PCR analysis of *DCAF1* KD efficiency, 300,000 HeLa cells were seeded in a 6-well plate and allowed to adhere overnight. The next day, cells were PEI-transfected with 300 ng of pcDNA mTag-BFP2 control plasmid, pcDNA mTag *DCAF1* KD shRNA #1 plasmid, or pcDNA mTag *DCAF1* KD shRNA #2 plasmid. Seventy-two hours post-transfection, cells were washed 1× with PBS, detached, and centrifuged at 500 RCF for 10 min. Cell pellets were resuspended in 500 μL TRizol reagent (Thermo Fisher, 15-596-018) and incubated at room temperature for 5 min. Next, 100 μL of chloroform was added, samples were vortexed and incubated at room temperature for 3 min prior to centrifugation at 12,000$g$ for 15 min at 4°. The upper phase was transferred to a fresh 1.5 mL Eppendorf tube and combined with 250 μL isopropanol for 2 hours at room temperature. Samples were centrifuged at 12,000$g$ for 15 min at 4°, pellets were washed 1× in ice cold 70% ethanol, and allowed to air dry. Pellets were resuspended in 20 μL RNase free $H_2O$ and immediately used for cDNA synthesis. For cDNA synthesis, 1.5 μg of RNA was incubated at 65 °C for 5 min in the presence of oligo dT primer. Then, RT buffer, dNTPs, RNase inhibitor, and reverse transcriptase were added to the reaction, incubated at 42 °C for 1 hour and then 70 °C for 10 min to inactivate reverse transcriptase. To amplify target genes, a 50 μL PCR reaction was performed following the reaction conditions described above using 0.5 μL of cDNA for 25 cycles.

## Supporting information

**S1 Data. Raw values used to generate bar graphs and scatter plots in the associated manuscript.** Data from a representative experiment are arranged in the accompanying excel sheet. Each figure panel is separated by individual tabs with the experimental sample names indicated above the raw values used to generate the respective figure panel.
(XLSX)

**S2 Data. Flow cytometry gating strategies.** Representative gating strategies are depicted for the indicated experiments. Accompanying figure panels for which the gating strategy is relevant are listed above each workflow.
(PDF)

**S1 Raw Images. Original immunoblots.** Original immunoblot images are depicted for the indicated figure panel with the dashed box representing the image used in the manuscript.
(PDF)

**S1 Fig. Vpr-induced global epigenetic remodeling in immortalized and primary MDMs.** (**A**) Representative fluorescence microscopy images of single nuclei from differentiated THP1 cells infected with control or Vpr-expressing viruses. Indicated histone marks are visualized using specific anti-acetylation or -phosphorylation antibodies. (**B**) Representative fluorescence microscopy images of histone marks in primary MDM cells infected with indicated viruses. (**C**) Quantification of histone marks following infection of primary MDM cells with Vpr$_{WT}$ or control viruses ($n = 75$ cells). The data underlying this Figure can be found in S1 Data.
(TIFF)

**S2 Fig. Vpr-induced global epigenetic remodeling in HeLa cells.** (**A**) Representative fluorescence microscopy images and quantification of histone marks in HeLa cells infected with indicated viruses ($n = 50$ cells). Analyses performed using a student $t$ test; ns, not significant; *** $p < 0.001$. The data underlying this Figure can be found in S1 Data. (**B**) Flow

cytometric analysis of DDR activation and histone marks in HeLa cells infected with indicated viruses. Representative gating strategies are depicted in S2 Data and raw FSC files can be found in the Figshare Data repository (https://doi.org/10.6084/m9.figshare.c.8239897). (**C**) Immunoblot analysis of histone marks in HeLa cells infected with indicated viruses. The unmodified images underlying this Figure can be found in S1 Raw Images. (**D, E**) Immunoblot analysis (D) and representative fluorescence microscopy images and quantification (E) of DDR activation in HeLa cells infected with indicated viruses ($n = 50$ cells). Analyses performed using a student $t$ test; *** $P < 0.001$. The data underlying this Figure can be found in S1 Data. The unmodified images underlying this Figure can be found in S1 Raw Images. (**F**) Quantification of DDR activation following infection of HeLa cells with the indicated virus in the presence or absence of vehicle, 10 nM $ATM_i$, or 10 mM $ATR_i$ ($n = 50$ cells). Analyses performed using a one-way ANOVA; ns, not significant; *** $p < 0.001$. The data underlying this Figure can be found in S1 Data.
(TIFF)

**S3 Fig. Vpr mutant-induced DDR and histone modifications.** (**A**) Amino acid alignment of $HIV_{NL4-3}$ and consensus subtype Vpr sequences. (**B**) Immunoblot analysis of indicated histone marks of $Vpr_{WT}$ infected HeLa cells in the presence and absence of 3 mM caffeine at 48 hours post-infection. (**C**) Quantification of histone marks following infection of HeLa cells with $Vpr_{WT}$ or mutant viruses ($n = 50$ cells). Analyses performed using a one-way ANOVA; ns, not significant; * $p < 0.05$; *** $p < 0.001$. The data underlying this Figure can be found in S1 Data. (**D**) Immunoblot analysis of Vpr protein expression in HeLa cells infected with the indicated viruses 48 hours post-infection. The unmodified images underlying this Figure can be found in S1 Raw Images. (**E**) Representative fluorescence microscopy images and RT-PCR analysis of DCAF1 expression in HeLa cells transfected with DCAF1 shRNAs. (**F**) Immunoblot analysis of fractionated HeLa cell lysates infected with indicated viruses 48 hours post-infection. The unmodified images underlying this Figure can be found in S1 Raw Images.
(TIFF)

**S4 Fig. Vpr modulation of HIV promoter activity.** (**A**) Flow cytometric analysis of HIV LTR activity in THP1 (top) and HeLa (bottom) cells infected with indicated viruses in the presence or absence of DDR inhibitors. Representative gating strategies are depicted in S2 Data and raw FSC files can be found in the Figshare Data repository (https://doi.org/10.6084/m9.figshare.c.8239897). (**B**) Quantification of total DNA-PK (DNPK) and phosphorylated DNA-PK (pDNPK) in HeLa cells infected with the indicated viruses in the presence or absence of ATM, ATR, and DNA-PK inhibitors ($n = 50$ cells). Analyses performed using a one-way ANOVA; ns, not significant; *** $p < 0.001$; ** $p < 0.01$. The data underlying this Figure can be found in S1 Data. (**C**) Quantification of DNPK activation in HeLa cells infected with $Vpr_{WT}$ or control viruses pre-treated with raltegravir or combination zidovudine/etravirine ($n = 50$ cells). Analyses performed using a one-way ANOVA; ns, not significant; *** $p < 0.001$. The data underlying this Figure can be found in S1 Data. (**D**) Quantification of gH2A.X foci in HeLa cells infected with the indicated viruses in the presence or absence of ATM, ATR, or DNA-PK inhibition ($n = 50$ cells). Analyses performed using a one-way ANOVA. ns, not significant. The data underlying this Figure can be found in S1 Data. (**E**) Quantification of mCh MFI of HeLa cells infected with $Vpr_{WT}$ or control viruses in the presence or absence of DNA-PK inhibition ($n = 4$ experiments). Analyses performed using a one-way ANOVA. *** $p < 0.001$; ** $p < 0.01$. The data underlying this Figure can be found in S1 Data. (**F**) Quantification of total RNA polymerase II ($RNA_{Pol}$) CTD (left), phosphorylated Ser2 (middle), and phosphorylated Ser5 (right) in HeLa cells infected with $Vpr_{WT}$ or control viruses in the presence or absence of ATM, ATR, or DNA-PK inhibition ($n = 50$ cells). Analyses performed using a one-way ANOVA; ns, not significant; * $p < 0.05$. The data underlying this Figure can be found in S1 Data. (**G**) Quantification of total SP1 (left), cJun (middle), and RNA polymerase II ($RNA_{Pol}$) CTD (right) in differentiated THP1 cells infected with $Vpr_{WT}$ or control viruses in the presence or absence of ATM, ATR, or DNA-PK inhibition ($n = 50$ cells). The data underlying this Figure can be found in S1 Data. (**H**) Immunoblot analysis of $RNA_{Pol}$ phosphorylation in HeLa cells infected with control or VprWT viruses in the presence and absence of DDR inhibitors. The unmodified images underlying this Figure can be found in

S1 Raw Images. (**I**) quantification of R-loop abundance in Hela and primary MDM cells infected with indicated viruses 48 hours post-infection ($n = 50$ cells). Analyses performed using a student $t$ test; *** $p < 0.001$. The data underlying this Figure can be found in S1 Data.
(TIFF)

## Author contributions

**Conceptualization:** Daniel J. Salamango.

**Formal analysis:** Nicholas Saladino, Emily Leavitt, Hoi Tong Wong, Jae-Hoon Ji, Diako Ebrahimi, Daniel J. Salamango.

**Funding acquisition:** Daniel J. Salamango.

**Investigation:** Nicholas Saladino, Emily Leavitt, Hoi Tong Wong, Jae-Hoon Ji, Diako Ebrahimi.

**Methodology:** Nicholas Saladino, Emily Leavitt, Hoi Tong Wong, Jae-Hoon Ji, Diako Ebrahimi.

**Project administration:** Daniel J. Salamango.

**Writing – original draft:** Nicholas Saladino, Daniel J. Salamango.

**Writing – review & editing:** Nicholas Saladino, Emily Leavitt, Hoi Tong Wong, Jae-Hoon Ji, Diako Ebrahimi, Daniel J. Salamango.

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
