## [Editor Report · Decision Letter 0]

7 Nov 2025

Dear Dr Salamango,

Thank you for submitting your manuscript entitled "HIV-induced DNA damage triggers epigenetic remodeling and transcription programs to enhance virus transcription and latency reactivation" for consideration as a Research Article by PLOS Biology.

Your manuscript has now been evaluated by the PLOS Biology editorial staff, as well as by an academic editor with relevant expertise, and I am writing to let you know that we would like to send your submission out for external peer review.

Once your full submission is complete, your paper will undergo a series of checks in preparation for peer review. After your manuscript has passed the checks it will be sent out for review. To provide the metadata for your submission, please Login to Editorial Manager (https://www.editorialmanager.com/pbiology) within two working days, i.e. by Nov 09 2025 11:59PM.

Kind regards,

Melissa

Melissa Vazquez Hernandez, Ph.D.

Associate Editor

PLOS Biology

---

## [Decision Letter · Decision Letter 1]

19 Dec 2025

Dear Dr Salamango,

Thank you for your patience while your manuscript "HIV-induced DNA damage triggers epigenetic remodeling and transcription programs to enhance virus transcription and latency reactivation" went through peer-review at PLOS Biology. Your manuscript has now been evaluated by the PLOS Biology editors, an Academic Editor with relevant expertise, and by two independent reviewers.

As you will see in the reports, all reviewers are positive about the relevance of the work, there were still some concerns and several suggestions for improvements. Reviewer 1 mentions concerns regarding clarity, mechanistic explanation, and data presentation, requiring improvement in figure quality, legends, labeling, and validation of Vpr mutant stability and subcellular localization. Reviewer 2 views the work as highly relevant and well executed, and raises minor points related to citation accuracy, figure legends, logical flow of DDR pathway introduction, and discussion of specific mutant phenotypes.

We expect to receive your revised manuscript within 1 month starting in January. Please email us (plosbiology@plos.org) if you have any questions or concerns, or would like to request an extension.

**IMPORTANT - SUBMITTING YOUR REVISION**

*Resubmission Checklist*

*Published Peer Review*

*PLOS Data Policy*

*Blot and Gel Data Policy*

I would like to take the chance to also wish you, your family and your team, the best on these coming holidays and a happy new year.

Sincerely,

Melissa

Melissa Vazquez Hernandez, Ph.D.

Associate Editor

PLOS Biology

REVIEWERS' COMMENTS

Reviewer #1:

Saladino et al. test the role of virion-associated and de novo Vpr in the induction of DDR responses, looking at epigenetic remodeling and activation of transcript by the LTR promoter of HIV-1 in acute infection and reactivation from latency. They find that Vpr exists as multimers in the nucleus and monomeric in the cytoplasm and that Vpr increases transcription from the HIV-1 LTR.

The paper reports a substantial body of research that addresses the still enigmatic function of the HIV-1 Vpr accessory protein. The experiments are carefully done with required controls and reference citations are accurate. The authors have developed some valuable tools in their studies, including the reporter viruses, Vpr mutants and split Venus fusion proteins.

The question of Vpr function has become quite complex, comprehensible only to those working directly on it. The studies are sophisticated but it will be hard for those outside the Vpr field to understand the story.

How does Vpr induce DDR? Presumably, it does not cleave the DNA. Is it the cell cycle arrest that does it? The authors should explain. They show that the DDR is mediated by both DECAF-dependent and independent mechanisms. How does Vpr mediate these two functions?

While there is a lot of data presented, it is important that it be easily intelligible to the reader. This should be improved throughout. The figures are tiny and the panels are squashed together, presumably to save space. They need to be made more easily visible. In addition, the figure legends do not adequately describe the experiments. It should not be necessary to consult M&Ms to understand the experiments.

Much of the data consists of histograms. Each bar shows a dot for each cell. Because of the large number of dots, the bars are difficult to discern. Why not remove the dots and substitute for error bars? Also, some of the labeling is difficult to understand, as detailed below.

Specific points

1. Fig1B. What do the greyed-out boxes represent? Are the genes deleted or just point-mutated?

2. Fig. 1 C. The histogram shows foci/cell or MFI. This was done using an EVOS 5000 imaging system. Some statement needs to be made of how the measurements were made. Does the microscope count the foci? Does it determine MFI/cell? The measurements were made presumably just of the mCherry+ cells. Is that correct? This needs to be stated in the figure legend.

3. Fig. 1D. the images are too hard to see and there is little explanation of what they are. This needs to be improved.

4. Fig. 2F. These images are too dark to see. This needs to be improved.

5. Fig. 2E. The viruses are mCherry and Vpr. This nomenclature is confusing. Don't both viruses have mCherry? Does the mCherry have a Vpr mutation? Labeling needs to be clarified to increase readability. Same for the other figures.

6. Fig. 3. This is a vast array of Vpr point mutants. Many Vpr point mutants are known to be unstable. A western blot is required to show the relative stability of all of the mutants.

7. Fig. 5E. The split venus experiment is good but the authors conclude that Vpr exists in 2 separate pools; one in the nucleus and one in the cytoplasm. The pool in the nucleus is clearly correct but the pool in the cytoplasm doesn't agree with the well-established finding that Vpr is a nuclear protein by immunofluorescence. How would it remain monomeric in the cytoplasm anyway?

8. Line 168. Change text to read "It is well established"

9. Line 183. Change text to induction of "H2A/.X foci".

Reviewer #2:

Saladino et al examine whether VPR alters the epigenomic landscape to promote increased HIV promoter activity. Here they present an extensive characterization of the changes to 17 histone marks in response to VPR+/- viruses in both HeLa and Thp1 cells. While they use microscopy as the primary assay, validation is also performed by western blot for key marks to confirm these changes via an alternate assay. Highly significant changes are also validated in primary macrophages. The authors use DDR pathway inhibitors to validate changes are linked to DNA damage induction and further perform a comprehensive mutational analysis of VPR to identify key interaction surfaces required for this activity. The authors examine the role of DCAF1 and DNA binding/nuclear retention in DDR in the generated mutants. Finally, the authors examine HIV promoter activity in an active infection model and latency models in response to VPR mutants and +/- DDR pathway inhibitors, demonstrating WT VPR does contribute to improved LTR activation during primary infection and reactivation from latency. The linkage of latency reversal to VPR is especially relevant and correlates with the known activity of HDACs as LRAs and recent work demonstrating HDACs can prevent latency establishment (https://pubmed.ncbi.nlm.nih.gov/38134881/). The increase in acetylation induced by both VPR and HDACs are likely both remodeling chromatin genome wide and at the LTR to promote activation, a link the authors do note in the discussion. This is a highly relevant paper that sheds light on the mechanisms of an enigmatic HIV protein. The work is well controlled, uses multiple assays and cell types for confirmation of observations, and is well presented in a logical flow.

Minor

1) Check citation 37 on page 4, related to DDR-induced histone PTMs associated with increased HIV transcription - this doesn't seem to fit.

2) Figure 2E - The legend does not mention primary macrophage data that is present, only references HeLa cells. It was also difficult to interpret that the histone marks were assessed by flow instead of microscopy. Would also recommend this be clarified in the text or legend.

3) Figure 2F - pCHK1/2 analysis is included here but the link between pCHK1/2 and DDR is not mentioned in the main text until Figure 3. Recommend moving this to introduce this earlier for those not immediately familiar with all DDR pathways.

4) Prior work has shown the S79A mutation (https://pmc.ncbi.nlm.nih.gov/articles/PMC11481869/) shows similar levels of H2A.x foci, however the S79E mutation presented here dramatically lowered these numbers. It would be nice to a comment about the differences between these mutations and potential hypothesis as to the different phenotypes in the discussion.

---

## [Editor Report · Decision Letter 2]

9 Jan 2026

Dear Dr Salamango,

Thank you for your patience while we considered your revised manuscript "HIV-induced DNA damage triggers epigenetic remodeling and transcription programs to enhance virus transcription and latency reactivation" for publication as a Research Article at PLOS Biology. This revised version of your manuscript has been evaluated by the PLOS Biology editors and the Academic Editor.

Based on our Academic Editor's assessment of your revision, we are likely to accept this manuscript for publication, provided you satisfactorily address the remaining editorial points. Please also make sure to address the following data and other policy-related requests.

1) We routinely suggest changes to titles to ensure maximum accessibility for a broad, non-specialist readership, and to ensure they reflect the contents of the paper. In this case, we would suggest a minor edit to the title, as follows. Please ensure you change both the manuscript file and the online submission system, as they need to match for final acceptance:

"DNA damage induced by HIV-1 Vpr triggers epigenetic remodeling and transcriptional programs to enhance virus transcription and latency reactivation"

2) Please add weblink of the funding agencies in the Financial Disclosure statement in the manuscript details.

Please supply the numerical values either in the a supplementary file or as a permanent DOI’d deposition for the following figures:

Figure: 1BCE, 2A-D, 3A-D, 4BCEF, 5ABC, 6CE, 7ABCDFGHI, 8ACDEF, S1C, S2AEF, S3C,S4B-GI

4) Please cite the location of the data clearly in all relevant main and supplementary Figure legends, e.g. “The data underlying this Figure can be found in S1 Data” or “The data underlying this Figure can be found in https://doi.org/10.5281/zenodo.XXXXX”

5) Please ensure that you are using best practice for statistical reporting and data presentation. These are our guidelines https://journals.plos.org/plosbiology/s/best-practices-in-research-reporting#loc-statistical-reporting and a useful resource on data presentation https://journals.plos.org/plosbiology/article?id=10.1371/journal.pbio.1002128

-- If you are reporting experiments where n ≤ 5, please plot each individual data point.

6) We require the original, uncropped and minimally adjusted images supporting all blot and gel results reported in the Figures 1AC, 4D, S2CD, S3BBEF, S4H

-- We will require these files before a manuscript can be accepted so please prepare and upload them now. Please carefully read our guidelines for how to prepare and upload this data: https://journals.plos.org/plosbiology/s/figures#loc-blot-and-gel-reporting-requirements

7) For figures containing FACS data (Figures 1D, 2D, 7DE, 8G, S2B, S4A), please provide the FCS files and a picture showing the successive plots and gates that were applied to the FCS files to generate the figure. We ask that you please deposit this data in an open repository like Zenodo and provide the accession number/URL of the deposition in the Data Availability Statement in the online submission form.

8) Supplementary files (e.g., excel). Please ensure that all data files are uploaded as 'Supporting Information' and are invariably referred to (in the manuscript, figure legends, and the Description field when uploading your files) using the following format verbatim: S1 Data, S2 Data, etc. Multiple panels of a single or even several figures can be included as multiple sheets in one excel file that is saved using exactly the following convention: S1_Data.xlsx (using an underscore).

9) Please add a scale bar in the following microscopy pictures in Figures: 6D

10) Please make sure that all figures use a colorblind-friendly palette.

11) Please ensure that your Data Statement in the submission system accurately describes where your data can be found and is in final format, as it will be published as written there

12) Per journal policy, if you have generated any custom code during the course of this investigation, please make it available without restrictions. Please ensure that the code is sufficiently well documented and reusable, and that your Data Statement in the Editorial Manager submission system accurately describes where your code can be found. More information on our Code Policy, what and how to share can be found here: https://journals.plos.org/plosbiology/s/code-availability

We expect to receive your revised manuscript within two weeks.

*Published Peer Review History*

*Press*

Sincerely,

Melissa

Melissa Vazquez Hernandez, Ph.D.

Associate Editor

PLOS Biology

---

## [Editor Report · Decision Letter 3]

14 Jan 2026

Dear Dr Salamango,

Thank you for the submission of your revised Research Article "DNA damage induced by HIV-1 Vpr triggers epigenetic remodeling and transcriptional programs to enhance virus transcription and latency reactivation" for publication in PLOS Biology. On behalf of my colleagues and the Academic Editor, Frank Kirchhoff, I am pleased to say that we can in principle accept your manuscript for publication, provided you address any remaining formatting and reporting issues. These will be detailed in an email you should receive within 2-3 business days from our colleagues in the journal operations team; no action is required from you until then. Please note that we will not be able to formally accept your manuscript and schedule it for publication until you have completed any requested changes.

PRESS

Sincerely,

Melissa

Melissa Vazquez Hernandez, Ph.D., Ph.D.

Associate Editor

PLOS Biology
